

# Bayesian data selection to quantify the value of data for landslide runout calibration

V Mithlesh Kumar[1], Anil Yildiz[1], and Julia Kowalski[1]

[1]Methods for Model-based Development in Computational Engineering, RWTH AACHEN UNIVERSITY, Aachen, 52062, Germany.

**Correspondence:** V Mithlesh Kumar (kumar@mbd.rwth-aachen.de) and Julia Kowalski (kowalski@mbd.rwth-aachen.de)

**Abstract.**

The reliability of physics-based landslide runout models depends on the effective calibration of its parameters, which are often conceptual and cannot be physically measured. Bayesian methods offer a robust framework to incorporate uncertainties in both model and observations into the calibration process. Therefore, they are increasingly used to calibrate physics-based

5    landslide runout models. However, the practical application of Bayesian methods to real-world landslide events depends on the availability and quality of observational data, which determines the reliability of the calibration outcomes. Despite this, systematic investigation of the influence of observational data on the Bayesian calibration of landslide runout models has been limited.

We propose quantifying the impact of observational data on calibration outcomes by measuring the information gained

10    during the calibration process using a decision-theoretic measure called Kullback-Leibler (KL) divergence. Building on this, we present a unified Bayesian data selection workflow to identify the most informative dataset for calibrating a given parameter. The workflow runs parallel calibration routines across available observation datasets. It then computes the information gained relative to the observations by calculating the KL divergence between prior and posterior distributions and selects the dataset that yields the highest KL divergence.

15    We demonstrate our workflow using an elementary landslide runout model, calibrating friction parameters with a diverse set of synthetic observations to evaluate the impact of data selection on parameter calibration. Specifically, we compare and quantify the information gained from calibration routines using observations with varying information content, i.e., velocity vs. position, and observations with different granularity, i.e., aggregated data vs. time series data. The insights from this study will optimize the use of available observations for calibration and guide the design of effective data acquisition strategies.





## 1 Introduction

Landslides are a significant natural hazard, with their frequency and intensity increasing as climate change makes extreme weather patterns more likely (Petley, 2012; Perkins, 2012; Wang et al., 2023). Understanding and predicting landslide runout behavior – how far and fast landslides travel (Xu et al., 2019) – is therefore crucial for impact-based risk and hazard assessment (Willenberg et al., 2009; Froese et al., 2012) and for the development of effective mitigation strategies (Mancarella and Hungr, 2010; Hübl et al., 2009). To this end, we use physics-based runout models, which are adept at capturing the bulk behavior of landslides, an essential objective in runout forecasting (McDougall, 2017). A wide variety of physics-based computational runout models are available, and they can be broadly classified on the basis of the fidelity level and flow characteristics accounted for, the underlying rheological relationships, and the computational methods used to solve them. McDougall (2017) and Trujillo-Vela et al. (2022) compiled a collection of selected computational models. Most of these landslide runout models are semi-empirical due to the absence of universal constitutive laws governing the complex and diverse phenomena involved in landslides (Pastor et al., 2012). Consequently, these models rely on empirical constitutive relations rather than mechanistic subscale formulations representing microscale behavior. This implies the presence of conceptual parameters (Iverson, 2003) that cannot be physically measured and thus must be calibrated based on the re-analyses of past landslide events. The applicability of such calibrated computational models is therefore limited to the physical regime covered by the available calibration data and offers little insight into the complex micromechanical properties of real landslides (McDougall, 2017). However, their ability to reproduce the bulk behavior of landslides makes them a pragmatic choice to analyze the landslide runout behavior (Hungr, 1995) and allows their use as operational tools in hazard mitigation.

Computational calibration methods play an essential role in the model-based prediction tool chain, particularly in landslide runout modeling. In the past, landslide runout models were predominantly calibrated using deterministic methods, traditionally based on subjective trial and error choices (Hungr and McDougall, 2009). These deterministic methods are limited by equifinality and non-uniqueness issues (McMillan and Clark, 2009) and do not offer a robust framework to handle multiple uncertainties, for instance due to model and data errors, involved in the calibration process (Barros et al., 2009). In contrast, probabilistic methods effectively address these limitations by aiming for the parameter's probability distributions rather than a deterministic estimate. Bayesian methods represent initial parameter uncertainty with a prior distribution, which is updated based on observational data to yield a posterior distribution reflecting the reduced uncertainty. This process thus offers a comprehensive framework to explicitly handle the various uncertainties involved in calibrating computational models that include conceptual parameters.

Recently, Bayesian methods have attracted significant interest from the geohazard research community. For example, Fischer et al. (2020) and Heredia et al. (2020) calibrated the rheological parameters of a snow avalanche propagation model employing a Bayesian approach. Aaron et al. (2019) applied this approach to calibrate a landslide runout model and compared its performance with a deterministic parameter identification algorithm. Furthermore, Moretti et al. (2017) inverted landslide characteristics using seismic data within a Bayesian framework. One significant finding in these studies that is also known from other application fields is that applying Bayesian methods typically entails a significant computational burden because of



the large number of so-called forward computational model evaluations required. This high computational cost is the primary
limitation of Bayesian methods in their practical applications (Aaron, 2017; Brezzi et al., 2016). A recent trend in simulation
for computationally costly high-throughput tasks is, therefore, to train non-intrusive surrogate models that aim at substituting
the original simulation model with a fast-to-evaluate alternative that, for example, can be utilized for uncertainty quantification
(Yildiz et al., 2023). Zhao and Kowalski (2022) leveraged surrogates based on Gaussian process emulation to develop a compu-
tationally feasible Bayesian calibration workflow, while Navarro et al. (2018) adopted a similar approach using the polynomial
chaos expansion ansatz to build surrogates.

Although integrating fast-evaluating surrogates into the calibration workflow alleviates the computational burden associated
with Bayesian calibration methods, this addresses only one part of the challenge. Effective calibration essentially relies on
reducing the discrepancy between model predictions and observational data (Cotter, 2024; Aaron et al., 2022). Two important
factors contribute to this discrepancy: i) the potential inadequacy of the model to capture the real-world process to be predicted,
and ii) the innate measurement errors within the observational data. Model inadequacy and its impact on the efficacy of the
calibration process have been addressed, for example, in the work of Kennedy and O'Hagan (2002); Heo et al. (2015); Xu and
Valocchi (2015).

The measurement error in the observational data affects the uncertainty of the predicted landslide runout when those data are
integrated into the computational calibration method underlying the model-based prediction tool chain. Methods for handling
measurement-related uncertainty are well established. These typically involve incorporating a statistical noise model —such
as a Gaussian distribution — into the calibration process. Hyperparameters such as mean and standard deviation are assumed
heuristically or inferred along with the model parameters (Zhao and Kowalski, 2022; Heredia et al., 2020) from the data.
Several studies use this approach to assess the influence of uncertainty in the observational data on the calibration outcomes
(Aaron et al., 2019). A typical outcome is, hence, insight into the acceptable measurement error to guarantee a specific quality
of the calibration result, which in turn dictates the reliability of the computational model predictions.

However, it is seldom considered that different types of observational data, such as an outline of the area affected by a land-
slide versus a localized measurement of its deposition height, result in markedly different calibration results. This observation
holds even if identical Gaussian noise assumptions are being used. Such differences indicate that the choice of observational
data not only plays a critical role in shaping calibration outcomes but also constitutes a lever to improve the quality of calibra-
tion outcomes. This insight is echoed in the work of Zhao and Kowalski (2022), who found that remarkably localized spatial
data, such as maximum velocities or deposit heights, provided better constraints for friction coefficients than aggregated data
like deposit volume and impact area. Moretti et al. (2017) made similar observations and found that force time history data
was more adept at inverting the characteristics of the landslide than static data such as deposit area or runout distance. This
is a surprising result since it strongly seems to indicate that even if we are ultimately interested in predicting the impact area,
physics-based computational landslide models should not necessarily be calibrated only based on the impact area of past re-
sults, but should also take into consideration local information such as a measurement of the deposition height at a specific
location.



Any systematic investigation of this effect requires an approach that quantifies the value of concrete choices of observational data and lays the foundation for systematically assessing the value-add of specific choices of observational data on the calibration outcome and, eventually, the predictive quality of the computational prediction pipeline. Such methods are currently unavailable, yet would be highly relevant to the geohazard community since observational data are often sparse due to logistical and financial constraints. Gaining clearer insight into how different observations influence calibration outcomes could support more efficient use of available data and guide the design of smarter data acquisition strategies. Similar ideas have been followed in other fields. Kavetski et al. (2011) examined the impact of data resolution on the inference of hydrological model parameters using data from an experimental basin, while Li et al. (2010) and Cui et al. (2015) investigated the effect of dataset length on the calibration of the hydrological model in data-limited catchments. In building energy modeling, Heo et al. (2015) studied the role of data quantity and quality in Bayesian calibration of the EnergyPlus model (U.S. Department of Energy, 2024). However, even these studies qualitatively compare the calibration outcomes and do not attempt to quantify the impact, which would help us to optimize data acquisition.

We can assess the impact of observations on the Bayesian calibration outcome by examining the resulting posterior distributions since they represent the uncertainty reduced during calibration. While Zhao and Kowalski (2022) reported variation in calibration outcomes across observational datasets by qualitatively comparing the corresponding posterior distributions, Moretti et al. (2017) analyzed this variation using the modes of the posterior distributions. However, neither of these approaches captures the inherent pathway in which observations influence Bayesian calibration: updating prior beliefs with information to arrive at the posterior distribution. Therefore, measuring this information gained during calibration would be an ideal approach to quantify the impact of observational data on calibration outcomes. To this end, we employ the Kullback-Leibler (KL) divergence, an information-theoretic concept, to compare probability distributions. Specifically, it measures information lost in approximating a probability distribution relative to the true distribution. Thus, we quantify the information gained during calibration by measuring the KL divergence between the posterior and prior distributions. Now, we have a quantitative metric to represent the impact of observation on the calibration outcomes, which can be used to select the most informative dataset. However, computing KL divergence poses a computational challenge because it requires calculating integrals that include the intractable posterior distributions. A widely adopted approach to tackle this challenge involves Monte Carlo-based estimators, which are prone to high variance related issues. Consequently, we choose a novel universal divergence estimator proposed by Wang et al. (2009). This estimator offers robust estimates of KL divergence leveraging k-nearest-neighbor (k-NN) distances.

This study addresses the identified research gap: a lack of a method to select the observational data best suited for the Bayesian calibration of landslide runout models by a) proposing a methodological approach and b) introducing a computational framework that demonstrates its feasibility. The developed Bayesian data selection workflow, which we define as finding the most informative dataset to calibrate a given parameter, allows us to quantify the information gained during calibration. This workflow orchestrates multiple calibration routines across different observational datasets in parallel. It then quantifies the calibration performance using KL divergence, which can be systematically exploited to optimize the data acquisition. To our knowledge, it is the first time that such information theoretic concepts have been incorporated to compare and quantify the calibration performance of landslide runout models.





We will demonstrate the proficiency of our Bayesian data selection methodology based on the so-called lumped mass model representing an idealized landslide runout model. The primary objective of this work is to introduce and describe a novel methodology, namely a Bayesian data selection workflow for landslide runout models. Application to the lumped mass model will allow us to use this workflow to investigate the role of the type and scope of observational data in the calibration process. In order to further isolate the role of data selection from secondary effects, we will limit ourselves to synthetic data generated from simulating an idealized lumped mass model at preselected set of parameters. While we know the limitations of a lumped mass point model, it provides an ideal testbed to assess the value-add of optimized data selection, which constitutes an unused potential hidden in landslide runout prediction. The community can also use our results as a future reproducible benchmark case.

## 2 Methodology

This section outlines a novel approach to quantifying the information gain in an automated Bayesian data selection workflow. As a first step, it is necessary to define the statistical model that underpins the embedded Bayesian calibration task, before introducing the KL divergence as a metric for measuring calibration performance across multiple datasets. Finally, we detail the computational workflow used to apply this methodology.

### 2.1 Statistical model formulation considering measurement and model error

The computational model, referred to as $\mathcal{M}$, comprises a physics-based theoretical model and a solution algorithm. Both in conjunction allow predicting the relevant aspects of the landslide hazard mitigation task, for example, the length of the runout. The computational landslide model can hence be written as a *parameter-to-observable mapping* given by

$$\mathcal{M} : S \rightarrow D \tag{1}$$

Here, $s \in S$ contains all the parametrized information needed to initialize a specific landslide simulation scenario, for example, topographic information and initial mass distribution, while $d \in D$ denotes the concrete prediction that is being made, such as the runout length or the impact area. Typically, space $D$ also comprises the space of observables $y$ that can be measured in the field, such that we assume $y \in D$. In our case, $\mathcal{M}$ additionally depends on parameter $\theta \in \Theta$ that cannot be determined independently and thus must be calibrated based on field observations $y$. Note that $\theta$ can either represent a scalar parameter or refer to a tuple of parameters, such as a single or several friction parameters.

The primary objective of model calibration is to leverage observations $y$ in order to infer on optimal parameters $\theta$, such that for a given scenario $s$ the discrepancy between observations $y$ and model predictions $\mathcal{M}(s;\theta)$ is minimized:

Calibration task: Find $\theta$ such that $|\mathcal{M}(s;\theta) - y|$ is minimal $\tag{2}$

Note, that we did not yet specify the metric, in which this deviation is measured. The discrepancy can be attributed to two significant sources of uncertainties, as discussed in the seminal work of Kennedy and O'Hagan (2002). First, we have the



uncertainty resulting from the noise in the observations, referred to as measurement noise, denoted by $\varepsilon$. Second, we have the model inadequacy, $\delta(s;\theta)$, which results from uncertainty and error in the model formulation itself.

### 2.1.1 Measurement noise

Field observations and measurements are inevitably affected by errors and noise (Kennedy and O'Hagan, 2002; Oden, 2016), which arise due to inherent limitations in measurement processes. For instance, measuring the runout distance of a landslide is often plagued with uncertainties resulting from sensor accuracy and topography resolution. Let us assume that our measurement $y$ is subject to noise, such that we have to differentiate it from $y^r$, denoting the true - yet unknown - representation of the physical phenomenon we want to capture. There will always be a discrepancy between $y$ and $y^r$, referred to as the measurement error and denoted by $\varepsilon$:

$$\varepsilon := y^r - y \tag{3}$$

Observations can hence be expressed as a function of $y^r$ and the associated measurement noise, according to

$$y = y^r + \varepsilon, \tag{4}$$

Typically, the noise is not known, such that we need to assume an ansatz, which constitutes our noise model. The additive Gaussian noise model is one of the most common noise models, where $\varepsilon$ is considered to be a realization of a Gaussian distribution, often with zero mean and a covariance matrix $\Sigma$, i.e. $\varepsilon \sim \mathcal{N}(0, \Sigma)$.

### 2.1.2 Model Inadequacy

Model inadequacy denotes the inherent limitations of the computational model in replicating the true value $y^r$, due to the idealizing assumptions and theoretical simplifications that underlie the physics-based model formulation. A certain idealization is evident in the case of landslide modeling, where the process complexity cannot be fully resolved. Kennedy and O'Hagan (2002) addressed this discrepancy by explicitly incorporating a model inadequacy term into the statistical model. Following Oden (2016), we refer to the model inadequacy term as $\delta(s;\theta)$ and we have the relation

$$y^r - \mathcal{M}(s;\theta) = \delta(s;\theta) \tag{5}$$

### 2.1.3 Statistical model and focus of this work

Combining the noise model Equation (4) and model inadequacy Equation (5) yields the statistical model.

$$y - \mathcal{M}(s;\theta) = \delta(s;\theta) + \epsilon, \tag{6}$$



which states that the deviation between the model predictions and the observations results from the superposition of measurement and model errors. As illustrated in Figure 1, this statistical model establishes the relation between model predictions, observations, and reality. Thus, it is used in the Bayesian calibration framework to find the *most probable* set of parameters for a given model $\mathcal{M}$ and observation $y$.

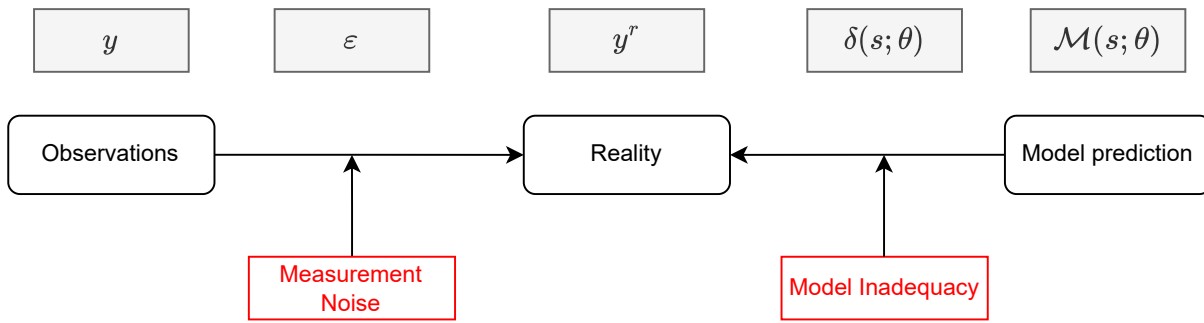

**Figure 1.** Statistical model connecting model predictions, reality, and observations. Adapted from Wu et al. (2018).

However, through this study, we aim to answer a different question: *Given a model parameter $\theta_i$, which is the most informative observation $y_i^*$ during calibration?* Consequently, we isolate the effect of data selection through the use of synthetic data generated from the computational model that guarantees observations to be consistent with the model predictions, hence

$$\delta(s;\theta) = y - \mathcal{M}(s;\theta) = 0$$

The resulting simplified statistical model can formally be re-written as

$$\theta = \mathcal{M}^{-1}(s;y,\epsilon). \tag{7}$$

In reality, of course, the situation is much more involved in the sense that $\mathcal{M}^{-1}$ denotes a computationally hard-to-solve calibration task. The formal write-up, however, indicates the aim of this study, which is not only to estimate parameter $\theta$ for a given observational dataset $y$ but also to interpret $y$ as a lever to improve the calibration result. The latter will be referred to as the **data selection** task. Developing a methodological approach to address this task is the major goal of this study. Although this goal differs from classical Bayesian calibration studies, it turns out that much of the existing work on Bayesian inference can be utilized.

## 2.2 Bayesian Inference framework

We adopt the Bayesian approach to solve the statistical model formulated in Equation (7) following Oden (2016). Underpinning everything that follows is Bayes' theorem, which reads



$$\underbrace{P(\theta \mid y)}_{\text{Posterior}} = \frac{\overbrace{P(y \mid \theta)}^{\text{Likelihood}} \cdot \overbrace{P(\theta)}^{\text{Prior}}}{\underbrace{\int_{\Theta} P(y \mid \theta) \cdot P(\theta)\, d\theta}_{\text{Evidence}}} \tag{8}$$

and captures the relation between prior knowledge of the parameter distribution for $\theta$, observation $y$, and information return after combining both. Bayes' theorem enables us to determine the optimal parameter values consistent with the observed data by computing the probability distribution of the parameters conditioned on the observations. The resulting distribution is referred to as the posterior distribution—or simply, the posterior—and is denoted by $P(\theta \mid y)$ in Equation (8). The posterior represents the updated beliefs about the parameters after observing the data. To arrive at this, we encode our prior beliefs regarding the parameters into a probability distribution known as the prior distribution, also known as the prior, denoted in Equation (8) as $P(\theta)$. We then update the prior distribution by multiplying it with the so-called likelihood function, $P(y \mid \theta)$, which, as the name implies, represents the likelihood of observing the data $y$ for a given set of parameters $\theta$. The prior distribution typically results from earlier Bayesian calibration steps or requires domain expertise or even empirical knowledge regarding the parameters. The likelihood function, on the other hand, follows from the employed noise model (Equation (7)) in the statistical model formulation. Assuming an additive Gaussian noise model with zero mean and covariance $\Sigma$, we formulate the likelihood function as shown below, where $N_d$ denotes the dimension of observation vector $y$.

$$P(y \mid \theta) = \frac{1}{\sqrt{(2\pi)^{N_d} det(\boldsymbol{\Sigma})}} \exp\left(\frac{-1}{2}(\mathbf{y} - \mathcal{M}(s;\theta)^T)\Sigma^{-1}(\mathbf{y} - \mathcal{M}(s;\theta))\right) \tag{9}$$

With the prior distribution and the likelihood function defined, we can now compute the posterior distribution using Bayes' theorem presented in Equation (8). However, computing the posterior distribution referred to as the evidence in Equation (8) involves integrating the product of likelihood and prior with respect to the parameter $\theta$. This integral is infeasible as soon as the computational model encapsulated in the likelihood function Equation (9) gets costly to solve. Also, the dimension of the parameter space $\Theta$ impacts on computational feasibility. For a high-dimensional parameter distribution, the integral is also high-dimensional, making it one of the primary practical challenges of the Bayesian approach. Following (Gelman et al., 2013; Robert and Casella, 2004), we address this challenge by approximating the posterior distribution using samples generated by Markov Chain Monte Carlo (MCMC) methods, averting the need to calculate the integral. These MCMC samples can be used to approximate the posterior distribution using methods such as kernel density estimation.

A schematic representation of the Bayesian approach is shown in Figure 2. We begin by defining the prior distributions for the parameters $\theta \in \{\theta_1, \theta_2, \ldots, \theta_m\}$. Then, the likelihood function is used to update these priors, thereby obtaining the posterior distributions. As Figure 2 illustrates, the likelihood function can be interpreted as the core driver of the process, with data resulting from observations guiding and refining each step. Consequently, the posterior distributions strongly depend on the observations used for calibration. The next section will be devoted to quantifying this dependence, ultimately leading to a better understanding of how to leverage observations $y$ for more informative and reliable posterior distributions.





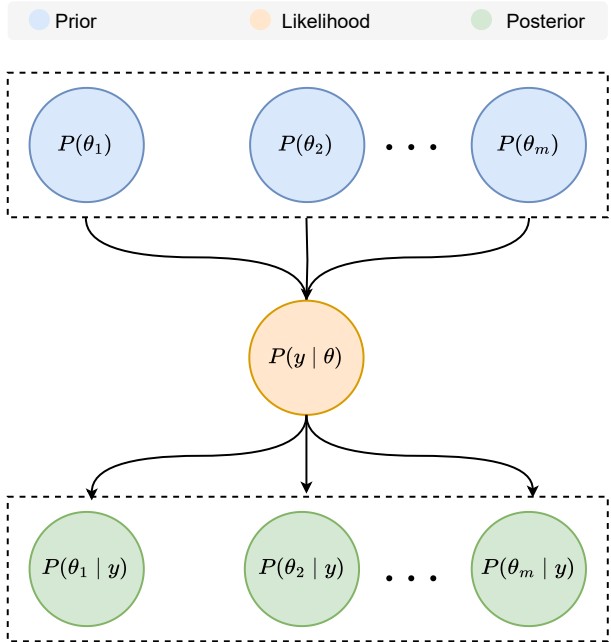

**Figure 2.** Schematic illustration of the Bayesian inference process. The likelihood function acts as the core driver of updating prior belief into posterior distribution of the parameter. Its outcome depends strongly on type and quality of the observation.

## 2.3 Data selection using Kullback-Leibler Divergence

While the Bayesian framework updates our beliefs about parameters $\theta$, the extent of this update depends on the specific observations used. To examine and quantify this influence, we measure the information gained when calibrating with a given observation $y$. For this purpose, we employ KL divergence (Kullback and Leibler, 1951), an information-theoretic measure of the dissimilarity between two probability distributions. Specifically, we evaluate the KL divergence between the posterior distribution ($P(\theta \mid y)$) and the prior distribution ($P(\theta)$) (Marzouk et al., 2007) as given below, which is then used for data selection.

$$D_{KL}(P(\theta \mid y) \,\|\, P(\theta)) = \int P(\theta \mid y) \log \left( \frac{P(\theta \mid y)}{P(\theta)} \right) d\theta \tag{10}$$

A schematic representation of the data selection process is shown in Figure 3. We use the Bayesian calibration approach detailed in Section 2.2 to calibrate the parameters $\theta$ using multiple observational datasets, $y_i \in \{y_1, y_2, ..., y_n\}$. Consequently, we obtain $n$ posterior distributions, one for each observational dataset, which we compare against the prior distribution using the KL divergence defined in Equation (10). This yields a quantitative measure of the information that each observation contributes to the calibration process, allowing us to identify the most informative dataset $\bar{y}^*$ to infer the parameters $\theta$. The bar above $\bar{y}^*$





indicates its aggregate nature. Although $\bar{y}^*$ maximizes the information for joint calibration, it may not be the most informative

dataset to calibrate a specific parameter $\theta_i$.

To identify the most informative dataset for a given parameter $\theta_i$, we marginalize the posteriors with respect to parameters and obtain $n$ posteriors for each parameter $\theta_i \in \{\theta_1, \theta_2, ..., \theta_m\}$, given as $\{P(\theta_i \mid y_1), P(\theta_i \mid y_2), ..., P(\theta_i \mid y_n)\}$. We then compute the KL divergence between the individual prior distribution $P(\theta_i)$ and the posterior distributions of $\theta_i$ corresponding to each of the observations. The generalized formulation for computing the KL divergence between the prior and posterior

distribution of the parameter $\theta_i$ calibrated using observation $y_j$ is given in Equation (11).

$$D_{KL}^{i,j}(P(\theta_i \mid y_j) \,\|\, P(\theta_i)) = \int P(\theta_i \mid y_j) \log \left( \frac{P(\theta_i \mid y_j)}{P(\theta_i)} \right) d\theta_i \tag{11}$$

Thus, for each parameter $\theta_i$, we have $n$ KL divergence values denoted as $D_{KL}^{i,j}$. These values correspond to observations $y_j$, with $j$ ranging from $\{1 \dots n\}$. Since these values quantify the information gained through the Bayesian update, we can utilize

them to select the dataset $y_i^*$ that constitutes the most informative dataset for calibrating the parameter $\theta_i$.

Data selection task: For a given parameter $\theta_i$ find $y_i^* = \arg \max_{j \in \{1, ..., n\}} D_{KL}^{i,j}$ $\qquad$ (12)

Bayesian calibration combined with data selection based on KL divergence constitutes the methodology of Bayesian data selection. This allows us to understand and quantify how adept available observations are in constraining a given parameter.

## 2.4 Workflow

We implement our Bayesian data selection workflow, as illustrated in Figure 5, using PSimPy, a Python-based package for predictive and probabilistic simulations (Zhao, 2022). The workflow comprises three distinct phases: (i) Surrogate Modeling; (ii) Bayesian Parameter Calibration; (iii) Data Selection.

### 2.4.1 Surrogate Modeling

Surrogate modeling serves as a computational enabler to overcome computational bottlenecks in calculating the KL divergence.

As illustrated in Equation (10), this calculation requires posterior distributions, which are approximated through MCMC sampling, as detailed in Section 2.2. The accuracy of this approximation is based on the ability of MCMC chains to explore the posterior space, which typically requires a large number of samples. Since drawing samples involves evaluating the computational model, this approach becomes infeasible for computationally expensive models. We therefore substitute the expensive computational model $\mathcal{M}$ with a cost-effective surrogate model through Gaussian process emulation, $\hat{\mathcal{M}}$. To achieve this, we

utilize the `emulator` module of PSimPy, which harnesses RobustGaSP, an R package for Gaussian stochastic process emulation that provides robust estimates of the hyperparameters leading to enhanced predictive performance. Furthermore, this





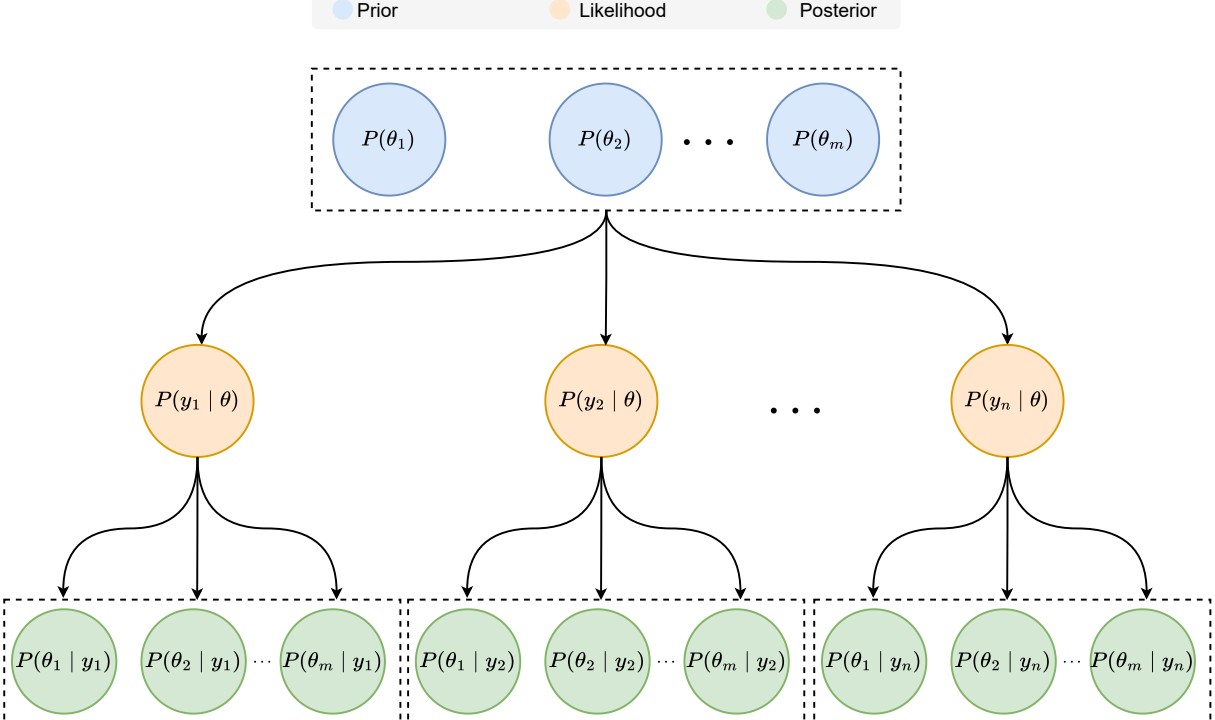

**Figure 3.** Schematic illustration of the Bayesian data selection process using KL divergence. Multiple calibration routines are performed in parallel, each using a different likelihood function to update the prior distributions. These likelihood functions correspond to the available observations and thus by comparing the resulting posterior distributions against the prior distribution we can quantify the information gained during calibration relative to observations.

package offers an efficient solution for handling multidimensional output cases using the parallel partial Gaussian stochastic process (PPGaSP) approach discussed in Gu and Berger (2016).

Figure 5 illustrates the key steps involved in the surrogate modeling phase. We start by generating a set of input parameters
using the `sampler` module of PSimPy, which leverages space-filling schemes like Latin hypercube sampling. Next, we employ the `simulator` module to evaluate our computational model ($\mathcal{M}$), at these input points. The resulting model outputs and the set of input points constitute our training data for the Gaussian process emulator. We used these training data to build and train the Gaussian process surrogate model. We then validate the trained surrogate using k-fold cross-validation. After successful validation, the surrogate model is available for predictions.

### 2.4.2 Bayesian Parameter calibration

The trained Gaussian process surrogate from the surrogate modeling phase replaces the expensive computational model in the likelihood function in Equation (9), allowing us to sample the posterior distribution using MCMC methods. For this purpose, we utilize the `mcmc sampler` of PSimPy, which is based on Python's `emcee` package, an affine invariant MCMC ensemble




sampler (Foreman-Mackey et al., 2013). This implies that the sampler is unaffected by affine transformations of parameter
space, allowing it to sample complex probability distributions without the need for problem-specific tuning. Furthermore, it
employs multiple chains that evolve in parallel, resulting in an efficient exploration of the probability distribution. Additionally,
we assess the convergence of the MCMC chains using the `diagnostics` module, leveraging Python's `Arviz` package
(Kumar et al., 2019).

### 2.4.3   Data selection

In this phase, we compute the KL divergence between the posterior and prior distributions obtained from the Bayesian cali-
bration phase. As discussed in Section 2.3, this involves computing the intractable integral, which we approximate using the
distance-based KL divergence estimator proposed by Wang et al. (2009). To this end, we incorporate the code from Hartland
(2020) into our workflow, as it implements the estimators described by Wang et al. (2009). To select the most informative
observation $y_i^*$ for a given parameter $\theta_i$, we marginalize the posterior distributions from the Bayesian calibration phase and
compute KL divergence, as presented in Section 2.3. Thus, if we calibrate $m$ parameters using $n$ observations, we end up with
$m \times n$ KL divergence matrix, as shown in Figure 4, where each entry quantifies the information that $j^{th}$ observation $y_j$ provides
for calibrating $i^{th}$ parameter $\theta_i$.

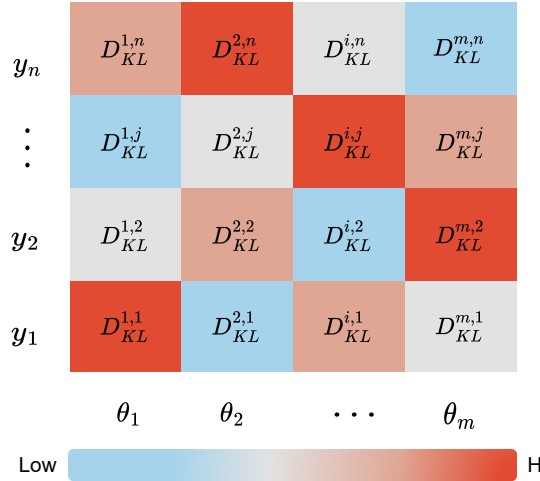

**Figure 4.** KL divergence matrix generated in the calibration of $m$ parameters with $n$ observation. Each entry of the matrix quantifies the
information provided by observation $y_j$ for calibrating parameter $\theta_i$.





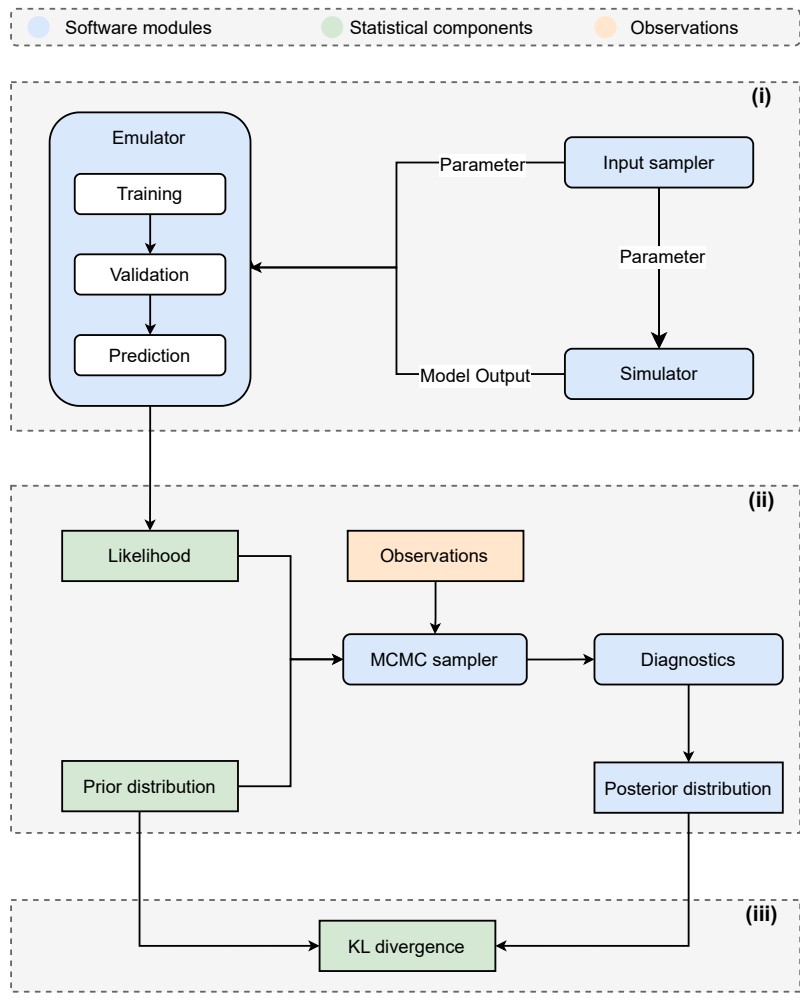

**Figure 5.** Overview of the Bayesian data selection workflow. The workflow consists of three key phases: (i) Surrogate Modeling, where a computationally efficient surrogate replaces the expensive forward model; (ii) Bayesian Parameter Calibration, where an MCMC sampler is used to infer the posterior distribution of the parameters; and (iii) Data Selection, where KL divergence quantifies the information gain from different observational data.



## 3 Case Study

This study investigates how the choice of observational data influences the calibration of model parameters. Specifically, we aim to identify the most informative dataset for calibrating a given parameter. To this end, we performed multiple calibration routines using the Bayesian calibration workflow outlined in Section 2.4, each with a different observational dataset. We then compute the KL divergence between the prior and posterior distributions to quantify the information gained from each dataset. As shown in Section 2.1, this process requires three key components: (1) a computational model with parameters to be calibrated, (2) observations to update those parameters, and (3) a noise model that captures uncertainty in observations. The noise model has already been described in Equation (4). This section describes the computational model and multiple observational datasets used in the calibration.

### 3.1 Computational Model

Hungr (1995) categorized the numerical landslide runout models into models based on continuum mechanics and lumped mass models. In both models, gravity primarily drives the motion, while a friction term, dependent on the chosen rheological model, resists it. However, due to the complexity of landslide dynamics, these rheological formulations often involve conceptual parameters that are not directly measurable and must, therefore, be inferred through model calibration. While lumped mass models idealize the sliding landslide mass as a single mass point, continuum-based models treat it as a spatially distributed deformable mass governed by conservation laws (Yildiz et al., 2023). As a result, lumped mass models are limited in their ability to represent internal deformations, which are captured by continuum-based models, thereby enabling a more accurate simulation of flow dynamics and deposit morphology (Hergarten, 2024). However, lumped mass models offer a conceptually straightforward framework for estimating bulk characteristics such as runout distances, velocities, and accelerations (Zahra, 2010). Furthermore, conceptual simplicity allows for a clear, tractable mapping between model parameters and landslide dynamics, which is often obscured in continuum models. Thus, we deliberately employ lumped mass models, given that the primary aim of this study is to assess how observational data influence parameter calibration.

Governing equation of a lumped mass model is mathematically described using Newton's second law of motion, as shown in Equation (13).

$$\frac{du}{dt} = g\sin\theta - \frac{F_{res}}{m} \tag{13}$$

Here, $u$ is the tangential velocity of the idealized mass point, $g$ is the gravitational constant, and $F_{res}$ is the resisting force term. We choose the Voellmy rheological model, which includes a classical dry Coulomb friction coefficient $\mu$ representing the basal resistance to landslide motion, along with a velocity-dependent friction term known as the turbulent friction coefficient $\xi$. The corresponding formulation of the resistance force is given as:

$$F_{res} = \mu mg\cos\theta + \frac{mg}{\xi}u^2 \tag{14}$$



In addition to the rheological parameters described in Equation (14) ($\{\mu,\xi\}$), we need to provide parameterized information ($s \in \{T(x,y), u_{x_0}, u_{y_0}, x_0, y_0\}$) specifying the landslide simulation scenario, where $T(x,y)$ denotes the topography
and the other parameters represent the initial conditions, such as initial velocity and position.

## 3.2 Observational datasets

We curate a diverse set of observations, each capturing different aspects of landslide dynamics, to systematically assess the influence of data selection on Bayesian calibration outcome. However, obtaining such diverse observational datasets in the real world is often infeasible due to logistical and financial constraints (Seibert et al., 2024). To address this, the study uses synthetic
data to calibrate the friction parameters of the lumped mass model described in Section 3.1. Synthetic data also provide the advantage of known ground-truth parameters that can be directly compared with the inferred values.

The observational data used in this study can be categorized as: (1) aggregate observations, which reflect bulk characteristics such as runout distance and maximum velocity, and (2) time series observations, which capture the velocity and position of the sliding mass over time. We generate these observations by evaluating the lumped mass model with a selected set of friction
parameters. These parameters are arbitrarily selected within the bounds reported in the literature: the dry Coulomb friction coefficient $\mu = 0.23$, chosen from the range $[0.02, 0.3]$, and the turbulent friction coefficient $\xi = 1000$, chosen from the range $[100, 2200]$. The lumped mass model is then evaluated with these parameters, generating a time history of velocity and position of the sliding mass. These outputs are post-processed to create both aggregate and time series observational datasets, with added noise to mimic real-world measurement uncertainty. For aggregate observations, which are scalar quantities (e.g., runout
distance or maximum velocity), perturbations are drawn from a Gaussian distribution with zero mean and a standard deviation equal to one-tenth of the scalar's magnitude. For time series observations, we assume that errors in time steps are independent of each other. Then, each time step is perturbed using noise drawn from a Gaussian distribution with zero mean and unit standard deviation and then scaled by one-tenth of the maximum value in the respective time series (velocity or position). The resulting noisy values are clipped to zero wherever they become negative.

## 3.3 Design of Numerical Experiments

We conducted a series of eight numerical experiments using the synthetic datasets described in Section 3.2 to calibrate the friction parameters of the lumped mass model. In experiments 1 through 4, we vary the information content and granularity of the observational data to examine the influence of data selection on the calibration outcome. Experiments 1 and 2 use aggregated observations—maximum velocity and runout distance, respectively—while Experiments 3 and 4 use time series
data of velocity $u(t)$ and position $x(t)$.

Experiments 5 and 6 investigate how the temporal characteristics of time series data, specifically length, and resolution, impact calibration outcomes. While experiment 5 involves calibration routines performed using time series data of varying lengths, experiment 6 uses time series data of varying resolution. Together, these experiments evaluate how the length and resolution of the time series data influence the accuracy of parameter estimation.



Experiments 7 and 8, extend the calibration to include the noise model's discrepancy parameter (see Equation (4)), which was previously fixed using heuristic assumptions. These experiments explore how jointly estimating observational uncertainty, along with $\mu$ and $\xi$, affects calibration outcomes. Velocity and position time series are again used as observational datasets in these cases.

**Table 1.** Summary of numerical experiments investigating the impact of observational data choice on Bayesian calibration outcomes. Experiments 1-7 calibrate friction coefficients ($\mu$, $\xi$) using different observation types, while experiments 8-9 additionally calibrate discrepancy parameters ($\sigma_{vel_{TS}}$, $\sigma_{pos_{TS}}$). Apart from experiments 5-6, all other experiments involve single calibration routines. All experiments use synthetic observations generated by adding random noise drawn from a Gaussian distribution with zero mean and specified standard deviations.

| Exp. No. | No. of Calibrations | Calibrated Parameters | Observation | Std. Dev. of Applied Noise |
|---|---|---|---|---|
| 1 | 1 | $\mu, \xi$ | Maximum velocity $U_{\max}$ | 2.08 |
| 2 | 1 | $\mu, \xi$ | runout distance $X_{\mathrm{end}}$ | 225 |
| 3 | 1 | $\mu, \xi$ | Velocity time series $u(t)$ | 2.08 |
| 4 | 1 | $\mu, \xi$ | Position time series $x(t)$ | 225 |
| 5 | 100 | $\mu, \xi$ | Multiple velocity time series, varying lengths | 2.08 |
| 6 | 10 | $\mu, \xi$ | Multiple velocity time series, varying frequencies | 2.08 |
| 7 | 1 | $\mu, \xi, \sigma_{vel_{TS}}$ | Velocity time series $u(t)$ | 2.08 |
| 8 | 1 | $\mu, \xi, \sigma_{pos_{TS}}$ | Position time series $x(t)$ | 225 |





## 4  Results

This section includes results for the curated set of numerical experiments discussed in Section 3.3. We adopt the workflow presented in Section 2.4 and the associated data: (i) Training dataset including set of sampled parameters and the corresponding model outputs; (ii) Ground-truth data used to generate the synthetic observations detailed in Section 3.2 are archived in this repository https://doi.org/10.5281/zenodo.17120721 (Kumar, 2025) .

### 4.1  Calibration using maximum velocity as observation

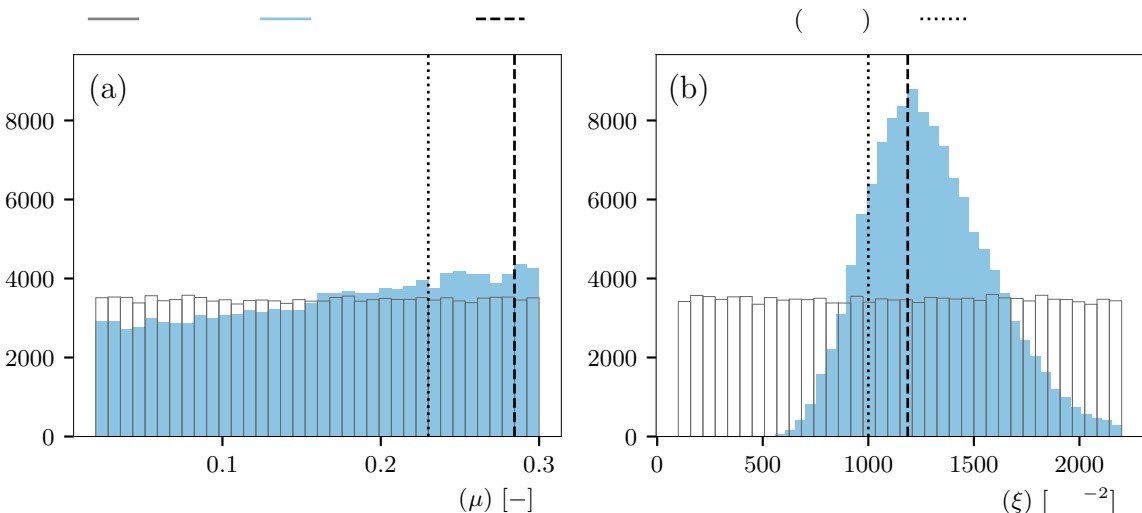

**Figure 6.** Posterior and prior distributions of (a) Dry Coulomb friction coefficient ($\mu$) and (b) Turbulent friction coefficient ($\xi$) based on calibration with maximum velocity ($U_{\text{max}}$) as observation. Maximum velocity conveyed more information about $\xi$ during calibration, as reflected in the narrower posterior distribution.

Figure 6 illustrates the prior distribution of the friction parameters and their corresponding posterior distributions obtained by performing the calibration using the maximum velocity ($U_{max}$) as observation. The maximum a posteriori (MAP) estimates, indicated by the black dashed lines in Figure 6a and Figure 6b, provide the most probable values for the parameters. The MAP estimate for $\mu$ is 0.284, while for $\xi$ it is 1186.73. The uncertainty associated with these estimates is summarized by highest density intervals (HDI), listed in the Table 2. The parameter values within $95\%$ HDI have a higher probability than

those outside this interval. Thus, a narrower HDI, as observed for $\xi$, indicates a lower degree of uncertainty. This suggests that the maximum velocity provided greater information for $\xi$ during calibration. This is also highlighted by the difference in KL divergence values, which had a higher value for $\xi$.





## 4.2 Calibration using runout distance as observation

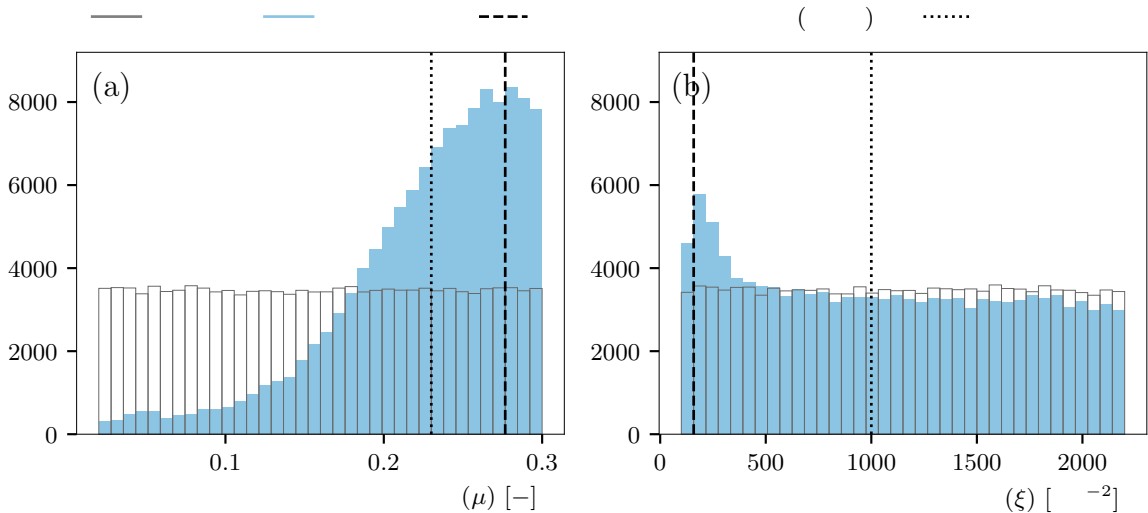

**Figure 7.** Posterior and prior distributions of (a) Dry Coulomb friction coefficient ($\mu$) and (b) Turbulent friction coefficient ($\xi$) based on a calibration with runout distance ($X_{\text{end}}$) as observation. Greater contraction of the posterior of $\mu$ indicates higher information gain during calibration with the runout distance.

We repeated the calibration of the friction parameters using the runout distance as observation, and the results are depicted
in Figure 7. This calibration yielded a MAP estimate for $\mu$ of 0.277, nearly identical to the MAP value presented in Figure 6a. However, the 95% HDI listed in Table 2, is considerably shorter, suggesting greater confidence in the estimate. In contrast, the 95% HDI interval for $\xi$ has widened, and even the MAP estimate (shown in Figure 7b) of 158.66 significantly differs from the true value of 1000. This indicates that more information was gained for $\mu$ when calibrated with the runout distance. A higher KL divergence value of 0.43 for $\mu$ compared to 0.05 for $\xi$ further corroborates this assertion, refer Table 2.



## 4.3 Calibration using velocity time series as observation

A third calibration was conducted using the velocity time series u(t), and the parameter distributions are presented in Figure 8. Unlike the results of calibration with aggregated data (e.g., maximum velocity and runout distance), we see a significant information gain for both $\mu$ and $\xi$, reflected by the much smaller $95\%$ HDI listed in Table 2. Even the KL divergence values outlined in Table 2 are significantly higher than those corresponding to the aggregated data. Furthermore, the MAP estimates for both parameters are nearly identical to the true values.

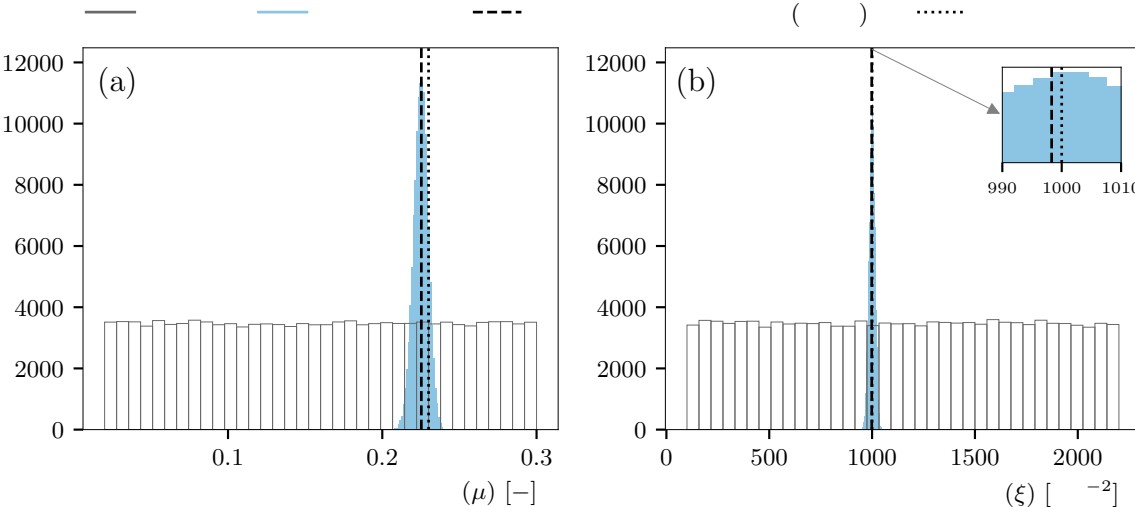

**Figure 8.** Posterior and prior distributions of (a) Dry Coulomb friction coefficient ($\mu$) and (b) Turbulent friction coefficient ($\xi$) based on a calibration with velocity time series as observation. Velocity time series provides considerably greater information about both $\mu$ and $\xi$ during calibration, as evidenced by the narrower posterior distributions.





## 4.4 Calibration using position time series as observation

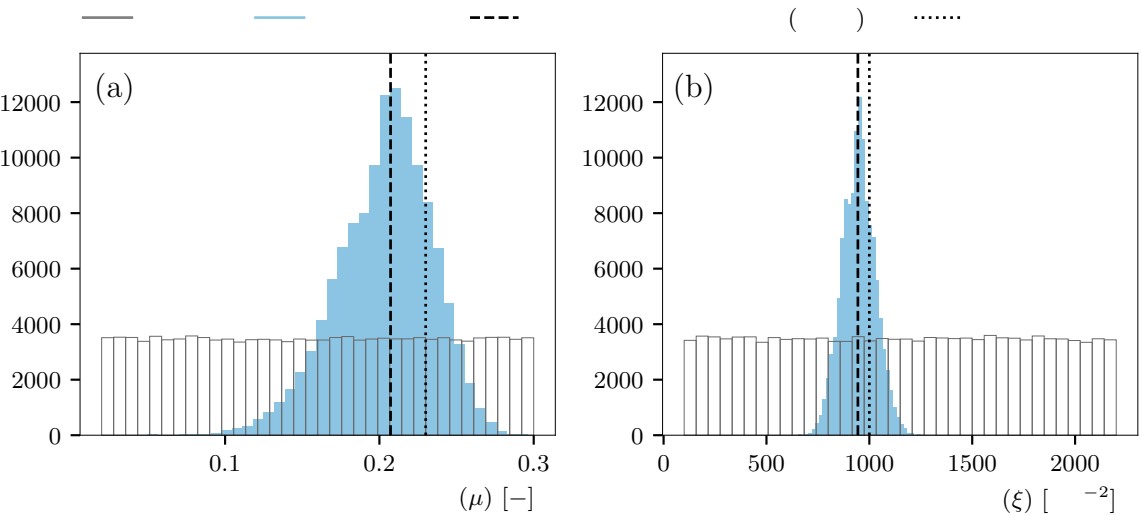

**Figure 9.** Posterior and prior distributions of (a) Dry Coulomb friction coefficient ($\mu$) and (b) Turbulent friction coefficient ($\xi$) based on calibration with position time series as observation. Similar to velocity time series, position time series also provides information about both parameters, albeit to a lesser extent.

Figure 9 shows the parameter distributions, calibrated with position time series, x(t). Again, we observed information gain for both parameters, reflected in the contracted posterior distributions, illustrated in Figure 9a and Figure 9b. MAP estimates for $\mu$ and $\xi$ are also closer to the true values compared to the estimates in Figure 6 and Figure 7. However, this information
gain is lower than that observed in the calibration using velocity time series ($u(t)$), as indicated by the lower KL divergence values of 0.84 for $\mu$ and 1.89 for $\xi$, compared to 2.69 and 3.59 (as shown in Table 2).



## 4.5 Value of information in data: Length of velocity time series data

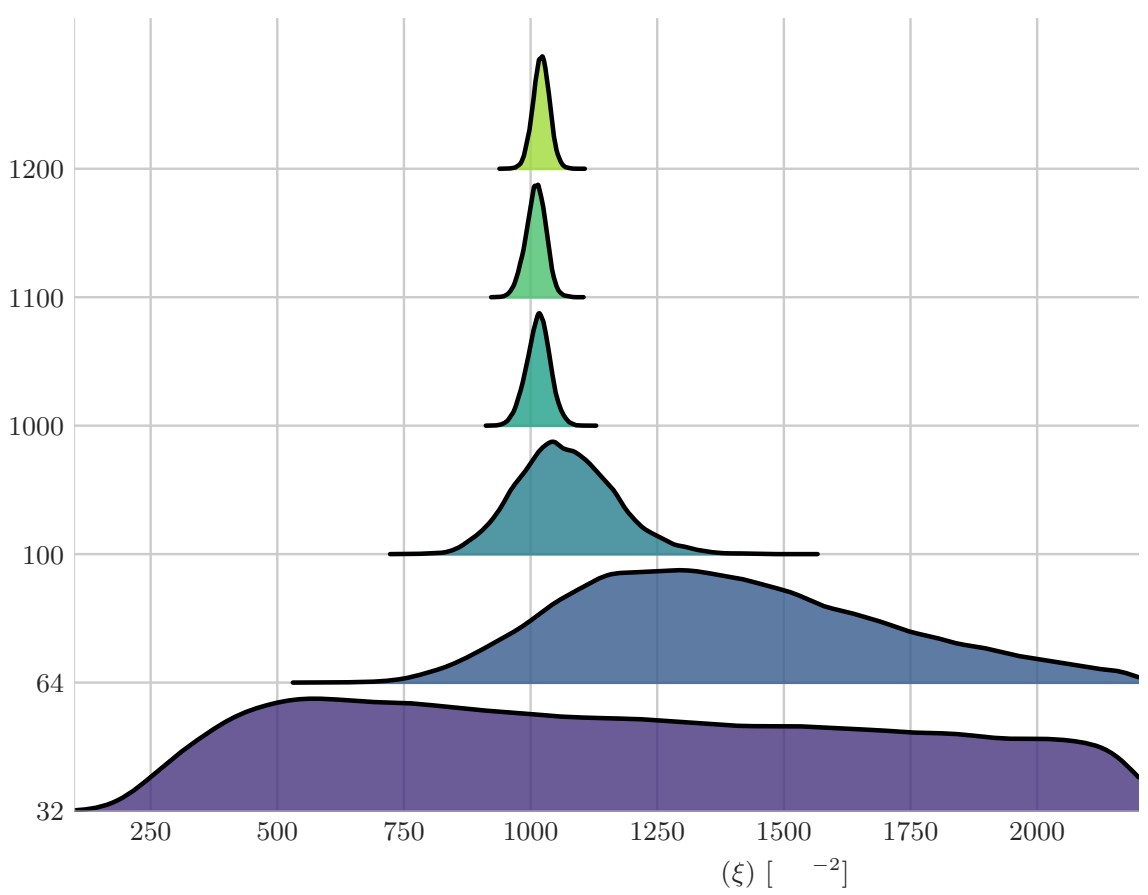

**Figure 10.** Posterior distributions of turbulent friction coefficient $\xi$ calibrated with a varying number of velocity time steps. The rate of information gain is more profound in the initial time steps than the later.

We calibrated the friction parameters with multiple datasets, each containing a varying number of velocity time steps. Selected posterior distribution of $\xi$ from these calibrations is depicted in Figure 10, with the y-axis listing the number of velocity time steps used in the calibration. Calibrating with a higher number of time steps led to greater information gain, reflected in the contracting posteriors with increasing time step count. However, from the Figure 10, we can see that the difference between the posteriors in the initial time steps is considerably greater than the posteriors in the later time steps. This suggests that the rate of information gain decreases as we use more time steps for calibration.

Figure 11a, plots the variation of the KL divergence of the posterior and prior distributions of $\xi$ with the velocity time steps used for calibration. Increasing time steps resulted in higher KL divergence, indicating a positive correlation between the information gain and the number of time steps. However, the initial steep slope of the plot in Figure 11a and its subsequent





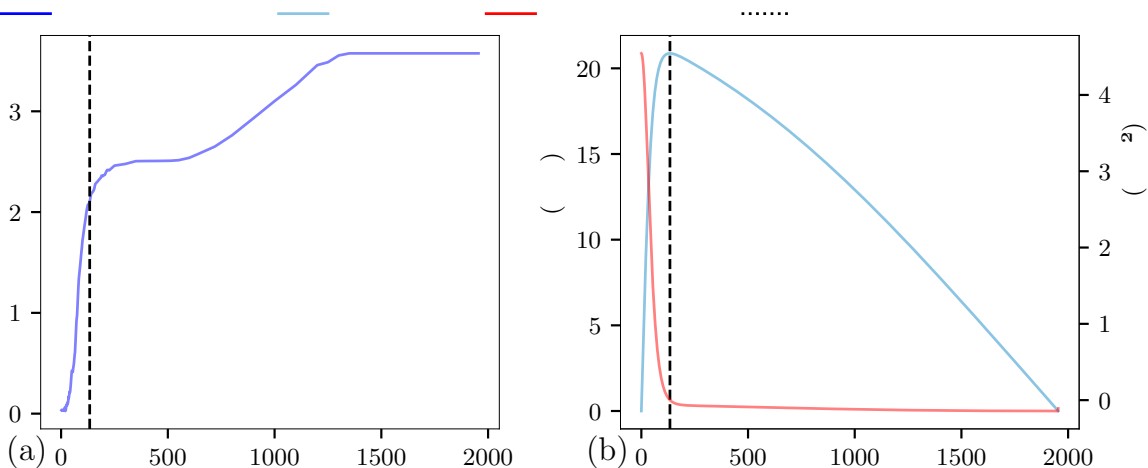

**Figure 11.** (a) Variation of Kullback–Leibler divergence of the posterior and prior distributions of turbulent friction coefficient $\xi$ with velocity time steps. (b) Variation of velocity and acceleration with time steps.

plateauing point to a diminishing return effect, where additional time steps beyond a critical threshold yield progressively smaller information gain. Interestingly, this threshold corresponds to the time step at which velocity attains its peak, as seen in Figure 11b.

**Table 2.** Kullback-Lieber Divergence and 95% Highest Density Intervals for friction coefficients calibrated using multiple datasets .

| Observations | Dry Coulomb Friction coefficient ($\mu$) | | Turbulent Friction coefficient ($\xi$) | |
|---|---|---|---|---|
| | KL Divergence | 95% HDI | KL Divergence | 95% HDI |
| Maximum Velocity | 0.06 | $[0.04, 0.3]$ | 0.63 | $[768.36, 1876.19]$ |
| runout distance | 0.43 | $[0.11, 0.3]$ | 0.05 | $[100.36, 2079.93]$ |
| Velocity Time Series | 2.69 | $[0.22, 0.23]$ | 3.59 | $[971.38, 1028.1]$ |
| Position Time Series | 0.84 | $[0.14, 0.26]$ | 1.89 | $[792.06, 1105.25]$ |





**4.6 Value of information in data: Temporal resolution of velocity time series data**

Figure 12 illustrates the posterior distributions of $\xi$ calibrated using velocity time series datasets of varying temporal resolutions. Each posterior distribution is associated with a specific time step size indicated on the y-axis. For example, a time step size of 4 on the y-axis indicates that the corresponding posterior was generated using a velocity time series of time step size 4. As the resolution of the time series data increases (i.e., with a smaller time step size), the information gain increases, as
evidenced by the contracting posteriors.

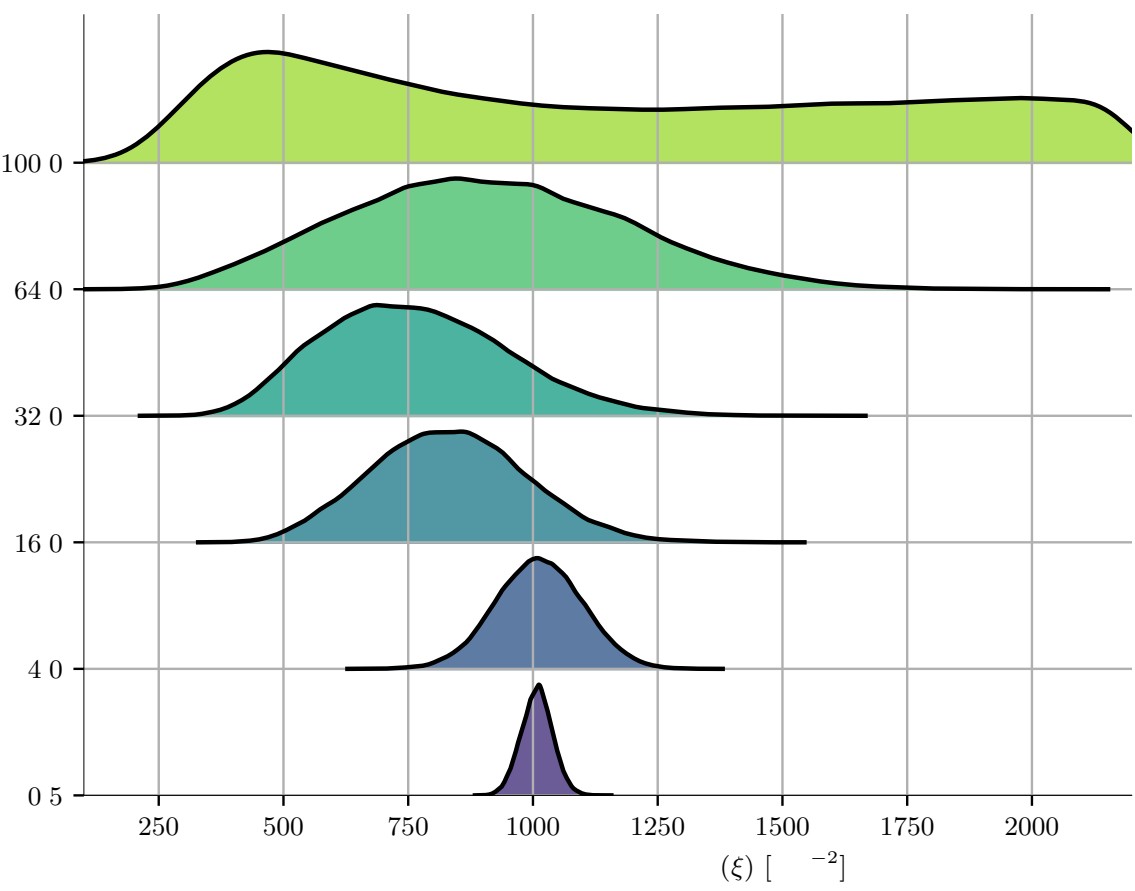

**Figure 12.** Posterior distributions of turbulent friction coefficient $\xi$ calibrated with a varying temporal resolution of the velocity data. The information gained during calibration increases with temporal resolution.



## 4.7 Calibration of the discrepancy parameters

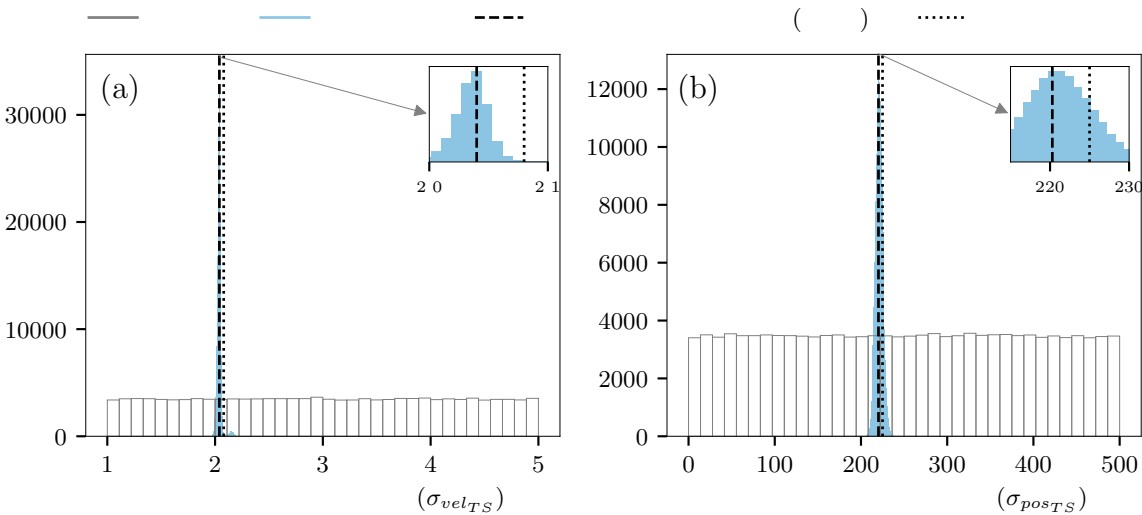

**Figure 13.** Posterior and prior distributions of (a) Velocity discrepancy parameter ($\sigma_{vel_{TS}}$) and (b) Position discrepancy parameter ($\sigma_{pos_{TS}}$) based on a calibration with velocity and position time series respectively.

Figures 6 to 9, presented posterior distributions of friction parameters calibrated using multiple datasets listed in Table 2. During these calibrations, we made heuristic assumptions for the discrepancy parameter $\sigma$ of the noise model, refer to the section 2.1 for detailed discussion. However, we can include this parameter in the calibration routine and calibrate it along with the friction parameters. Figure 13, depicts the prior and posterior distributions of discrepancy parameters corresponding to velocity and position time series. In Figure 13a, we see that using velocity time series $u(t)$, we can learn extensively for the velocity discrepancy parameter $\sigma_{vel_{TS}}$, indicated by the highly contracted posterior. Similarly, the position time series provides substantial information regarding the position discrepancy parameter $\sigma_{pos_{TS}}$, refer Figure 13b.



## 5 Discussions

In our numerical experiments, we investigated the impact of the scope and quantity of the data used as observations in the Bayesian parameter calibration of landslide runout models. To this end, we calibrated the friction parameters of a lumped mass model, namely the dry Coulomb friction coefficient ($\mu$) and the turbulent friction coefficient ($\xi$), with a diverse set of observational data summarized in Table 1. Using our novel Bayesian data selection workflow, we quantified the information gained during the calibration by means of a statistical distance measure from information theory, referred to as KL divergence

(Kullback and Leibler, 1951; Marzouk et al., 2007). Comparison of KL divergence associated with posterior distributions of alternative observation data (Figures 6 to 9) highlights the critical role of data selection in the calibration of these parameters. In our experiment, we found that calibration of a lumped mass model using maximum velocity and runout distance revealed contrasting trends: the maximum velocity provided substantial information for $\xi$ while minimally contributing to $\mu$. In contrast, calibration based on the runout distance provided a greater constraint, hence more information, for $\mu$, but minimally

contributed to gaining a better understanding of $\xi$. These contrasting trends can be explained by examining the variations of these parameters with maximum velocity and runout distance, respectively. Figure 14 shows a clear dependence of the runout distance on $\mu$, which decreases with increasing $\mu$ while remaining unaffected by changes in $\xi$. In contrast, the maximum velocity varies significantly with $\xi$ but shows little to no change with $\mu$. The narrower posterior distributions for $\xi$ when calibrating with maximum velocity and for $\mu$ when calibrating with the runout distance highlight the complementary strengths of these

datasets for Bayesian parameter inference. This behavior is consistent with earlier findings of McDougall (2017), who noted that $\mu$ and $\xi$ influence different aspects of flow behavior. Specifically, $\mu$ is associated with the runout distance, while $\xi$ limits the flow velocities. Our results extend this qualitative understanding by providing a methodology to quantify these relationships using the information gained during Bayesian calibration. These outcomes point to the hidden potential of a systematic data selection regarding its anticipated value-add with the critical process governed by the parameter of interest.

In numerical experiments 1 and 2, aggregated data proved inadequate in the joint calibration of the parameters — they could infer only one parameter each — we, therefore, explored time series data as an alternative. We calibrated the parameters using velocity and position time series data, each depicting the time history of velocity and position of the sliding mass, respectively. Figures 8 and 9 illustrate that the time series data provided substantial information during the calibration of both parameters, as evidenced by the highly contracted posteriors. This higher information gain implies that time series data are better equipped

to calibrate friction parameters than aggregated data. Our findings are consistent with the work of Moretti et al. (2017), who observed that the time history data offered a better calibration of the landslide parameters than the static data. Specifically, the force-time history derived from seismic records was more adept at constraining landslide parameters than runout distance and deposit area because it captures the temporal evolution of landslide dynamics. In the same way, the velocity and position time series data used in our study captured the evolving dynamics of landslides more efficiently than aggregated data. A similar

observation was made by Yan et al. (2022), who emphasized the limitations of static data in the adequate inversion of the landslide characteristics. Additionally, the friction parameters we want to calibrate govern the acceleration and deceleration phases of the landslide motion, which are better reflected in the time series data (Moretti et al., 2017). These findings further




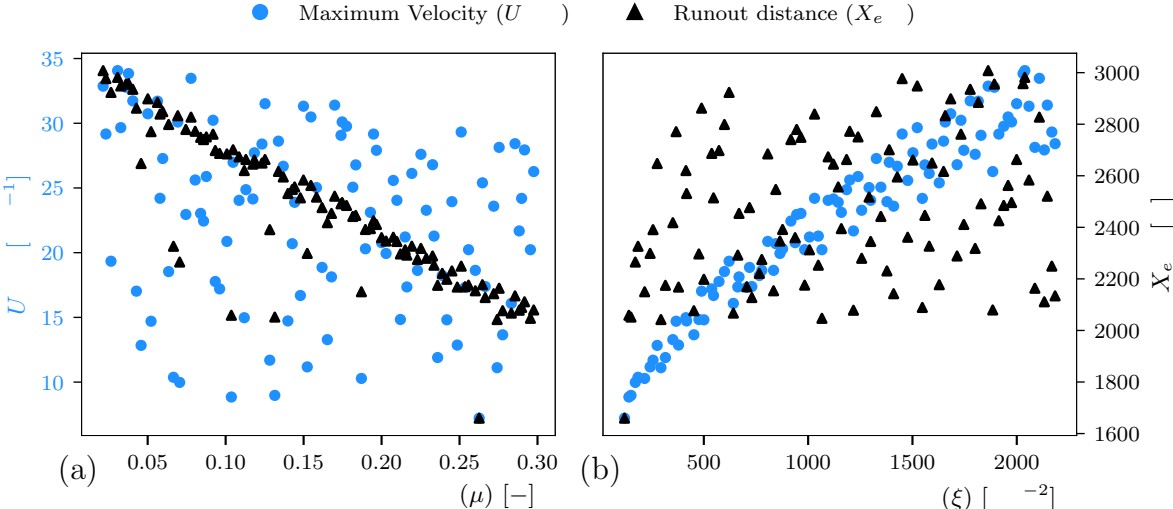

**Figure 14.** Variation of the friction coefficients ($\mu$ and $\xi$) with maximum velocity ($U_{\mathrm{max}}$) and run out distance ($X_{\mathrm{end}}$). The dry Coulomb friction coefficient $\mu$ primarily varies with $X_{\mathrm{end}}$, while the turbulent friction coefficient $\xi$ is more strongly influenced by $U_{\mathrm{max}}$.

suggest that aligning observations with the critical process governed by the parameter of interest improves the calibration performance (Kavetski et al., 2011).

The superior performance of time series data compared to aggregated data implies a positive correlation between data quantity and calibration performance, where data quantity refers to the length and resolution of the time series. From Figure 10, we can see that an increase in the length of time series leads to enhancement in calibration indicated by the contracting posteriors. Similarly, the calibration improves when we increase the temporal resolution, evidenced by posterior contraction with decreasing time step size, as shown in Figure 12. These observations further support the claim that as the data quantity

used for calibration increases, the calibration performance enhances accordingly. These observations echo findings in the field of hydrology, which highlight that increasing the data series length enhances the reliability of the calibration of a hydrological model(Cui et al., 2015; Li et al., 2010). Kavetski et al. (2011) reported similar findings; they compared the impact of the temporal resolution of the data on the calibration of the hydrological model parameters. They found that high-resolution data better captured parameters associated with fast hydrological processes, as these finer-scale data preserve the dynamics that are

lost in coarser resolutions because of data averaging. However, more data does not necessarily lead to better calibration; for instance, Ekmekcioğlu et al. (2022) found that increasing the length of calibration data series beyond 10 years did not improve the validation performance of the hydrological model. Similarly, Etter et al. (2018) observed that the ability of the data to capture critical processes related to the parameters we want to calibrate was more critical than the temporal resolution of the data.

As the practical availability of data is often limited due to logistical and financial constraints (Seibert et al., 2024), we analyzed the information gained relative to the data points in the time series data. For this study, we calibrated friction parameters





using multiple time series datasets, each containing several time steps. Figure 10 depicts the posterior distributions of the parameter $\xi$ calibrated using a selected set of time series data of varying length. From Figure 10, we can infer that as the number of time steps increases, the rate of information gain decreases, which implies that the ratio of information gain to the length of time series data is skewed after a certain threshold. This inference aligns with the study of Li et al. (2010), who found that the hydrological model calibration did not improve after a certain threshold, even with increased data series length. This inference is further reflected in Figure 11a, where we plotted the information gain (quantified by KL divergence) against time steps. We can observe that the slope of this plot flattens after a certain threshold, indicating again that the rate of information gain diminishes after a threshold. We further observed from Figure 11b that this threshold corresponds to an observation window in the velocity time series during which the sliding mass accelerates and attains its maximum velocity. Thus, this is the duration which marks the point at which the system's dynamics have evolved and stabilized, as critical processes affecting the dynamics have happened. Therefore, data capturing these changes is significantly more informative and relevant than the rest. This finding reinforces our earlier point: it is not solely the data quantity that matters, but rather its ability to capture the critical processes governing the parameters.

Figure 13a and Figure 13b indicate that the discrepancy parameter is successfully calibrated using velocity and position time series. In this study, since we used synthetic data for calibration, we had control over the noise in the data; refer Section 2.1 and Table 1 for details. Specifically, we know the standard deviation of the Gaussian distribution from which the noise was drawn; for velocity time series, it was 2.08, and for position time series 225, see Section 3.2. These values closely align with the MAP estimates of the discrepancy parameter from the calibration of the velocity and position time series of 2.03 and 220, indicating our ability to infer this parameter and, thus, reflecting our ability to quantify the uncertainty associated with measurement noise. These findings further indicate that higher data quantity (like time series data) can help us quantify the uncertainty associated with the data quality. Our results are consistent with Khorashadi Zadeh et al. (2019), who reported that higher data quantity can offset the impact of poor data quality.

## 6 Conclusions

The outcome of Bayesian calibration is highly dependent on the available observational data. Despite this well-known fact, there is a lack of studies that systematically investigate the impact of the choice of observation on the calibration result. We propose a Bayesian data selection workflow to address this challenge and identify the most informative observation in calibrating a given parameter. This workflow quantifies the impact of data selection on calibration performance by assessing the information gained during calibration, utilizing KL divergence, an established information-theoretic metric. Computing the KL divergence based on posterior distributions resulting from Bayesian parameter calibration presents itself as an extremely computationally intense task. We addressed the latter challenge by integrating a surrogate modeling technique based on Gaussian process emulation. The complete Bayesian data selection workflow is being made available with this article.

We have demonstrated the feasibility of the workflow through numerical experiments in which we systematically investigated the influence of data selection in calibrating two friction parameters of an idealized landslide runout model. To achieve this, we




designed rigorous experiments that quantitatively assess how observations with variations in information content, specifically velocity versus position—and granularity, such as aggregated data versus time series data—affect the calibration outcome. The experimental results indicate that the information content and the observation granularity significantly impacted the calibration outcome. We found that time series data considerably outperforms aggregated data in constraining parameters, owing to its superior ability to capture the landslide dynamics. However, this does not imply that calibration performance scales linearly with the data quantity. While increasing the length and frequency of time series data enhances calibration performance, these improvements yield diminishing returns if the observation window exceeds a specific duration. Remarkably, the optimal length of the observation window seems to correspond to the time the sliding mass needs to attain its maximum velocity. Thus, the evolution of landslide dynamics has stabilized. These findings suggest that data capturing the specific dynamics for an observation window of that duration are better suited to calibrate landslide model parameters. The landslide community can use these insights to optimize calibration strategies based on available data and to design effective future data acquisition strategies.



*Code availability.* Code required to perform the numerical experiments listed in Section 4 is available at this repository https://doi.org/10.5281/zenodo.17120721 (Kumar, 2025).

*Data availability.* Data required for the experiments: (i) Posterior samples corresponding to the calibration routines, (ii) Training data comprising of the design (sampled set of input parameters) and the corresponding model outputs (iii) Ground truth data used in the calibration routines is hosted in this repository https://doi.org/10.5281/zenodo.17120721 (Kumar, 2025).

*Author contributions.* Author contribution following the CRediT taxonomy. V Mithlesh Kumar: Conceptualization, Methodology, Software, Formal analysis, Investigation, Writing—original draft, Writing—review and editing, Visualization. Anil Yildiz : Conceptualization, Methodology, Writing—review and editing, Supervision, Visualization. Julia Kowalski : Conceptualization, Methodology, Writing—Review and Editing, Supervision, Funding acquisition.

*Competing interests.* The authors declare no competing interest relevant to the content of this article.

*Financial support.* This research has been supported by financial support from DFG-Deutsche Forschungsgemeinschaft (German Research Foundation) under the grant 333849990/GRK2379 (International Research Training Group (IRTG-2379): Hierarchical and Hybrid Approaches in Modern Inverse Problems).



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
