# Peer review of "Bayesian data selection to quantify the value of data for landslide runout calibration"

_EGUsphere, 2025_

## Referee Comment (RC1)

**Review of manuscript**

**Bayesian data selection to quantify the value of data for landslide runout calibration by V Mithlesh Kumar, Anil Yildiz, and Julia Kowalski**

The authors present a framework for the friction and measurement noise parameter estimation in landslide observations. These values are essential for landslide runout models in order to obtain reliable values for runout length and velocity. That makes the work very valuable and beneficial. They use a Bayesian approach to estimate the information gain based on various observations. The amount of information gain is measured by the Kullback-Leibler-Divergence of the prior and posterior parameter distributions in the Bayesian inference. Based on the KL divergence the most relevant observations for calibration of a certain parameter is chosen. This is a promising approach that has the potential to improve landslide runout predictions.

**General remarks:**

The first question that comes to my mind is if the results are easily transferable from one region to another. Once a landslide occurred and observations for the calibration are available I suppose the danger of a landslide at the same location is rather low (my understanding, I am not a specialist in this field). If so what are the values of the information gathered at one site? Can they be used at other landslide-threatened locations? So, the applicability of the results should be pointed out a bit better.

There is no description of the geometry (or anything else) of the landslide model the authors used to generate the synthetic data for the experiments. This way it is hard to estimate if it is a very idealised case that is difficult to transfer to reality or not. Also, the friction parameters in the model are uniform on the entire sliding surface. How would the results look like if the sliding surface was complicated and parameters vary within the model?

How did you choose the prior distributions for the friction and discrepancy parameters? They look almost evenly distributed, but not quite. Why?

Regarding information gain: KL divergence is just a measure for the dissimilarity of distributions. Its value could be high even though the resulting MAP is far from the true value. So it is a little bit misleading (or optimistic) to call it information gain. The authors need to elaborate a bit why the value of KL divergence alone lets one infer the improvement of the parameter estimation.

Almost all axis labels are missing in the figures. Also labels of curves in the legends. This makes reading (and reviewing) the manuscript rather inconvenient since some guess work is involved. This is a major mistake and must be fixed!

**Specific remarks:**

- 1. 43: "parameters" since rest of sentence is plural.
- 1. 65: better: "uncertainties"
- 1. 113: What does "high variance related issues" mean exactly?
- 1. 162 &164: the equations are not consistent. Exchange y^r and y in eq. (3)?

- l. 177: why does the appearance of epsilon change? Typo or different meaning? If the latter, explain!
- 1. 210: The operator det(\Sigma) should not be written in italic.
- 1. 218: Better describe MCMC method used. How is stationarity ensured?
- 1. 238: What is the structure of  $\{y\}^{*}$ ? Is it an element of  $\{y_1, y_2, ..., y_n\}$ ?
- 1. 255: Figure 5 is referenced before Figure 4.
- 1. 264: Further explain and justify usage of Gaussian emulator.
- 1. 270: PsimPy is written in typewriter-like font here but in normal font earlier. Please be consistent.
- l. 284: Section 2.4.3 (Data selection) describes the process of computing the KL divergence for all combinations of observations and parameters. However, nothing is written about selecting data (or observations) based on the resulting matrix shown in Fig. 4. This needs to be described.
- 1. 316: Subscripts that are not variables or indices should not be italic but normal font (i.e.
- F\_{\mathrm{res}}). Furthermore, it is inconvenient to use \theta for the angle here, since it is used as variable for the parameters earlier in the manuscript.
- 1. 360: Experiment numbers (1–9) do not match numbers in table.
- 1. 361: Italic subscripts that are not variables or indices.
- 1. 364, Figure 6: Some information are missing / not printed (at least in the provided pdf file): legend incomplete (no text next to lines);

no axis labels; unit of \xi not printed:

Same issues with all the other figures of prior and posterior distributions. When it is unclear which line represents which quantity, it is nearly impossible to estimate the information content and correctness of the figures! This especially difficult for figure 11.

- 1. 366: Subscript "max" should not be italic.
- 1. 368: Are six significant digits for \xi justified here?
- 1. 372: Where are the KL divergence values given / shown?  $\rightarrow$  reference Table 2
- 1. 387: "x(t)" here is just text, not typeset as a formula. Please be consistent.

- 1. 393: Figure 10: axis labels are missing. Are these examples taken from a set of 100 experiments? If so please state that fact. Otherwise the reader wonders how from 6 values you get such a smooth curve in figure 11a.
- 1. 396: How is it visible that initial time steps have greater information content? That does not become clear. You show graphs for varying numbers of time steps but do not compare different onsets of the time series. Please clarify!
- 1. 399: better: "number of velocity time steps". Otherwise it could be interpreted as KL divergence vs. specific time steps. Same in next sentence on line 400: better "Increasing number of time steps..."

Figure 11a: What is the reason for the shape of the KL div curve? Why is there a plateau between 300 and 500 steps and then an increase again?

I suppose figure 11 shows number of time steps used (a) and time steps (b) but that's a guess since x-axis labels are missing. If that is so the start of the plateau corresponds to about 200 steps. It would be interesting to see that length also in figure 10. In figure 10 the three uppermost curves (1000, 1100, 1200 steps) look very similar. In figure 11a, however, the KL divergence is still increasing significantly in that range. Why is that?

- 1. 410: Figure 12 is missing axis-labels again. A plot of KL divergence vs. time step would be informative, too. Which time window is used here for the observations?
- 1. 413/414: you refer to section 2.1 where the discrepancy, i.e. measurement error is called \epsilon. Here you write \sigma which seems to refer to the STD of applied noise as given in Table 1. Be consistent with naming scheme!
- 1. 512: Figure 11b shows the max. acceleration (assuming it is the blue line) at about 200 (no unit given). Figure 10 shows a narrow distribution only for 1000 steps or more. So how do you come to this optimality statement? That does not seem reasonable. There is not even a curve for 200 steps in figure 10.

---

## Author Comment (AC2)

**Revision letter: response to Reviewer #1**

**egusphere-2025-4531**
**Bayesian data selection to quantify the value of data for landslide runout calibration**

December 18, 2025

We would like to thank Reviewer #1 for their valuable comments and suggestions, which improved the quality of the manuscript substantially. We present herein our responses to the comments of Reviewer #1.

**Italicized** line, figure, section numbers refer **to the revised manuscript**.

The changes in the revised manuscript with respect to the original manuscript can be traced in the `diff.pdf` file. Note that the line numbers in this file deviate from the pure manuscript PDF due to the tracked changes displayed.
* * *
**Legend**

> Shows original reviewer comment

**Response** Contains direct response and explanations.

**Resulting changes in manuscript** Summarizes resulting changes in the paper.

> Contains a direct quote of new or modified manuscript content with significant changes with respect to the original manuscript.

**1 Comments of Reviewer #1**

**1.1 General comments**

**1.1.1 Comment 1**

> The first question that comes to my mind is if the results are easily transferable from one region to another. Once a landslide occurred and observations for the calibration are available I suppose the danger of a landslide at the same location is rather low (my understanding, I am not a specialist in this field). If so what are the values of the information gathered at one site? Can they be used at other landslide-threatened locations? So, the applicability of the results should be pointed out a bit better.

**Response**

Thank you for this comment.

- **Regarding transferability of calibration results:** You are correct that calibration results from one landslide site are generally not directly transferable to other locations. The landslide runout models often contain conceptual parameters, that cannot be measured physically, making site-specific calibration necessary. However, we note two important caveats: (i) repeated landslide events do occur at specific sites, where calibration results can inform the design of informative prior distributions, thus improving future calibrations [Zhao and Kowalski, 2022]; and (ii) Bayesian calibration aids in building databases of parameter distributions across sites, which can enable probabilistic forecasting at sites where landslides have not yet occurred [Aaron et al., 2022].

- **Regarding the applicability of results from this paper:** This is a methodological paper rather than an application study. Our primary contribution is a systematic framework for assessing how data selection impacts Bayesian calibration of landslide runout models, essentially quantifying the value of different observational datasets for parameter estimation. We deliberately idealized the problem (simplified topography, computational model, etc.) to isolate the effect of data selection on calibration outcomes. The practical value of this work lies in enabling a priori assessment of data informativeness before costly field campaigns and calibration efforts. Researchers can use our framework with their specific topographies, runout models, and synthetic data to: (i) identify which measurements are most informative for specific parameters, (ii) optimize data acquisition strategies, and (iii) select optimal subsets from existing observational datasets. For example, our finding that the observation window with maximum information gain rate aligns with the acceleration phase (when the mass reaches maximum velocity) suggests prioritizing observations of transient dynamics. Such insights, derived from scenario testing with synthetic data, can guide cost-effective monitoring strategies tailored to specific sites and objectives.

**Resulting changes in manuscript**

To address the reviewer feedback, we elaborated the applicability of the work in the discussion section (*p 29. lines 524-529*).

> Beyond the insights from our synthetic case study, the proposed Bayesian data selection workflow provides a systematic framework for performing a priori assessment of data informativeness before costly field campaigns. To implement this framework, practitioners should first prepare a synthetic test case using site-specific topography and their computational model. By generating candidate observational datasets through forward model evaluation at known parameters, they establish a virtual testbed for scenario-based testing that quantifies how different candidate datasets inform specific parameters of interest and tests case-specific hypotheses about data value.

**1.1.2 Comment 2**

> There is no description of the geometry (or anything else) of the landslide model the authors used to generate the synthetic data for the experiments. This way it is hard to estimate if it is a very idealized case that is difficult to transfer to reality or not. Also, the friction parameters in the model are uniform on the entire sliding surface. How would the results look like if the sliding surface was complicated and parameters vary within the model?

**Response**

Thank you for the comment.

- **Regarding the geometry of the landslide model:** We acknowledge that the details of geometry used in this study can be elaborated. We have added *Figure 6* to the revised manuscript, which is reproduced here as Figure 3 for convenience. The idealized synthetic topography consists of a curved longitudinal profile that is uniform in the transverse direction.

- **Regarding the transferability of results:** This study intentionally employs simplified topography to isolate the effect of data selection on calibration outcomes—the core methodological question we address. While this represents an idealization compared to real landslide sites, it provides a controlled environment for systematically investigating our methodological question.

  Importantly, the two key insights from our study are geometry-independent and founded in the fundamental mechanisms of interaction between model parameters and observations: (i) time series data provide better constraints than scalar data, and (ii) parameters are most effectively calibrated by observations that capture their dynamical influence. The idealized setup allows us to demonstrate these insights clearly; researchers can then apply the same framework to their site-specific topographies to identify analogous patterns relevant to their particular cases.

- **Regarding friction parameters:** Yes, the friction parameters are assumed constant over the entire sliding surface. This assumption is widely used in landslide runout modeling, as it captures the bulk behavior of landslides (runout distance, velocity) with a parsimonious model structure. While a complex model with friction parameters varying within the surface would be more physically accurate, it would also be difficult to calibrate. The available observations would not be adequate to constrain them. In summary, while we can formulate advanced models with spatially varying parameters, their efficacy in real-world applications is limited by the availability of adequate observations.

**Resulting changes in manuscript**

We have added the synthetic topography used in this study as an additional figure. Kindly refer to Section 3.1, (*p 18. Figure 6*).

**1.1.3 Comment 3**

> How did you choose the prior distributions for the friction and discrepancy parameters? They look almost evenly distributed, but not quite. Why?

**Response**

Thank you for the comment.

- **Regarding the choice of prior:** In this study, we assumed noninformative (uniform) prior distributions for both the friction and discrepancy parameters. Such uniform priors are commonly used in the literature for calibrating landslide runout models. We have included a description of the prior distributions and justified their choice with relevant citations in Section 3.3 (Design of Numerical Experiments).

- **Regarding the apparent unevenness:** This is a visual artifact of histogram binning used for plotting, not a characteristic of the underlying continuous uniform distributions. The priors are truly uniform (flat) over their specified ranges.

**Resulting changes in manuscript**

Added description of priors used in Section 3.3 (*p.17, lines 391-395*), repeated below for convenience.

> In all the 8 experiments we assume uniform priors for the friction parameters. The bounds for the priors are chosen from the literature as mentioned earlier, for $\mu \in [0.02, 0.3]$ and $\xi \in [100, 2200]$ $m/s^2$. While such uniform priors are noninformative, they are widely used in literature for calibrating landslide models (Aaron et al., 2019; Navarro et al., 2018; Moretti et al., 2017). Additionally in experiment 7 and 8, we assume an uniform prior for the discrepancy parameters $\sigma_{vel_{TS}}$ and $\sigma_{pos_{TS}}$ with bounds $[1, 5]$ and $[0, 100]$ respectively.

**1.1.4  Comment 4**

> Regarding information gain: KL divergence is just a measure for the dissimilarity of distributions. Its value could be high even though the resulting MAP is far from the true value. So it is a little bit misleading (or optimistic) to call it information gain. The authors need to elaborate a bit why the value of KL divergence alone lets one infer the improvement of the parameter estimation.

**Response**

Thank you for this comment.

- **Regarding KL divergence as a measure of information gain:** We agree that KL divergence is fundamentally a measure of distributional dissimilarity. However, in Bayesian inference, the KL divergence from posterior to prior has a specific information-theoretic interpretation: it quantifies the reduction in uncertainty about parameters achieved through observing data. This interpretation, where $D_{\mathrm{KL}(P_{\mathrm{Posterior}}||P_{\mathrm{prior}})}$, represents information gain, is standard in Bayesian optimal experimental design and inverse problems [Huber et al., 2023, Chowdhary et al., 2024]. Higher KL divergence indicates stronger posterior contraction, meaning observations have reduced parameter uncertainty and constitute information gain.

- **Regarding KL divergence alone as measure of improved calibration:** We agree that higher KL divergence alone does not inherently guarantee that posteriors are centered on true parameter values. As you correctly note, one could have high KL divergence with posteriors confidently concentrated around incorrect values, particularly in cases of model misspecification or strongly informative but misplaced priors.

  However, the synthetic experimental design used in this study specifically addresses these concerns and validates KL divergence as a measure of calibration performance in our context. First, we generate synthetic observations from the computational model with added noise, eliminating model misspecification. Second, we employ uniform (noninformative) priors over physically plausible parameter ranges, avoiding the risk of strong priors constraining posteriors away from true values. Under these conditions, posterior contraction (quantified by KL divergence) directly reflects movement toward true parameter values, not arbitrary concentration.

  This relationship between KL divergence and posterior accuracy is demonstrated empirically in our results. *Table 2* shows that calibration with velocity and position time series yields higher KL divergence values compared to calibration with maximum velocity and final position alone. Examining the corresponding posterior distributions in *Figures 9* and *10* (these were Figures 8 and 9 in the original manuscript), we observe that cases with higher KL divergence exhibit posteriors concentrated around the known true parameter values (indicated by dashed vertical lines). This confirms that in our experimental framework, higher KL divergence reliably indicates greater information gain and improved parameter estimation accuracy.

To summarize, our work follows the information-theoretic interpretation of KL divergence as a measure of information gain and it employs controlled conditions (synthetic data, noninformative priors) under which higher KL divergence reliably signals improved calibration. This framework can be extended by other researchers using synthetic data to systematically analyze how different observations influence parameter estimation in their specific contexts. However, when applying this approach in the presence of model misspecification, KL divergence must be complemented with additional validation methods such as the posterior predictive checks.

**Resulting changes in manuscript**

To address the reviewer feedback, we have expanded the definition of KL divergence in *Section 2.3 (p. 9, lines 236-242)* and included two supporting citations.

> In the Bayesian setting, KL divergence between the posterior distribution ($P(\theta \mid y)$) and the prior distribution ($P(\theta)$), as defined in Equation (10), admits a direct interpretation as information gain, since it quantifies the reduction in uncertainty about parameter $\theta$ achieved by conditioning on observation $y$. Higher KL divergence indicates stronger posterior contraction relative to the prior, implying that the observation provides greater information about the parameters; accordingly, we use this quantity as a criterion for data selection, consistent with its standard use in Bayesian optimal experimental design and inverse problems (Marzouk et al., 2007; Huber et al., 2023; Chowdhary et al., 2024).

Additionally we included the limitation on the use of KL divergence as measure of improved parameter calibration in the presence of model misspecification and prior-data conflict, (*p. 29, lines 534-537*).

> Second, when model misspecification or prior-data conflict occurs, posteriors can contract away from true parameter values, where higher KL divergence does not necessarily indicate better calibration. When applying this framework in such scenarios, practitioners should therefore complement KL divergence with robust validation methods such as posterior predictive checks.

**1.1.5 Comment 5**

> Almost all axis labels are missing in the figures. Also labels of curves in the legends. This makes reading (and reviewing) the manuscript rather inconvenient since some guess work is involved. This is a major mistake and must be fixed!

**Response**

Thank you for the comment. We acknowledge that when the manuscript was posted on the preprint server on 02.10.2025, it had rendering issues which resulted in several figures missing axis labels and legends. We apologize for this inconvenience. However as I already posted in the interactive forum as author comment this error was related to rendering issue and not with the submitted manuscript. As soon as we realized this issue on 06.10.2025 we communicated to the editorial office who promptly corrected it the same day. The properly rendered manuscript with all axis labels and legends has been available online since 06.10.2025. We have verified that all figures in the current online version are displayed correctly with complete labeling.

**Resulting changes in manuscript**

No changes to the manuscript content were necessary. The issue was resolved by correcting the rendering of the existing figures, which now display all axis labels, legends, and annotations as intended.

**1.2   Specific comments**

**1.2.1   Comment 1.43**

> "parameters'" since rest of sentence is plural.

**Response**

That is correct. It has been fixed.

**Resulting changes in manuscript**

The relevant sentence now has been corrected (*p.2 lines 43-44*).

**1.2.2   Comment 1.65**

> better: "uncertainties".

**Response**

Thank you for the suggestion. We have included it in the revised manuscript.

**Resulting changes in manuscript**

We have implemented the changes throughout the manuscript and in *Figure 1*.

**1.2.3   Comment 1.113**

> What does "high variance related issues" mean exactly?

**Response**

Thank you for the comment. In this sentence we are referring to the higher variance of the Monte Carlo-based estimators. Monte Carlo methods yield unbiased estimates of KL divergence but suffer from high variance due to their random sampling nature. Although KL divergence is always a positive quantity, the estimator can produce negative values for individual runs, and many samples are required for the random fluctuations to cancel out and reveal the true KL value.

**Resulting changes in manuscript**

We have revised the sentence to make the argument more specific and direct (*p.4, lines 112-113*).

**1.2.4   Comment 1.162 and 1.164**

> the equations are not consistent. Exchange $y^r$ and $y$ in eq. (3)?

**Response**

Thank you for catching that. We have corrected the equation accordingly.

**Resulting changes in manuscript**

We corrected *Equation 3* (*p.6, lines 163*), to be consistent.

**1.2.5   Comment 1.177**

> why does the appearance of epsilon change? Typo or different meaning? If the latter, explain!

**Response**

Thank you for catching that. It was a typo and we have corrected it.

**Resulting changes in manuscript**

We have corrected the typo in *Equation 6 (p.7, lines 182)*.

**1.2.6   Comment 1.210**

> The operator $det(\Sigma)$ should not be written in italic.

**Response**

Thank you for catching that. We fixed it.

**Resulting changes in manuscript**

We have corrected the typo in *Equation 9 (p.8, lines 216)*.

**1.2.7   Comment 1.218**

> Better describe MCMC method used. How is stationarity ensured?

**Response**

Thank you for pointing this out. While the general idea of MCMC sampling is introduced at line 218, the specific implementation details are located in *Section 2.4.2 (p.12-13, lines 306-313)*.

- **Regarding the MCMC method used:** The specific MCMC algorithm we used was the Affine Invariant Ensemble Sampler (via the Python `emcee` package). This sampler employs multiple chains (walkers) in parallel to efficiently explore the parameter space while remaining invariant to affine transformations of the parameter space, thereby avoiding the need for problem-specific tuning.

- **Regarding the stationarity of the MCMC chains:** Stationarity (convergence to the posterior distribution) was ensured and verified in two ways: (i) Burn-in removal: We discarded the initial 50% of samples as burn-in to eliminate the influence of starting positions; and (ii) Diagnostics: We quantitatively assessed convergence using the Gelman-Rubin statistic ($\hat{R}$) and autocorrelation time analysis using the `ArviZ` package, in addition to qualitative visual inspection of trace plots.

**Resulting changes in manuscript**

We expanded the explanation of the convergence of MCMC chains in *Section 2.4.2 (p.13 lines 311-312)* to include details on the convergence criteria.

**1.2.8 Comment 1.238**

> What is the structure of $\bar{y}^*$? Is it an element of $\{y_1, y_2, , y_n\}$?

**Response**

Yes, $\bar{y}^*$ belongs to the set of candidate datasets, $\{y_1, y_2, \ldots y_n\}$. Thanks for bringing this to our attention. We realized that the notation related to the observational datasets could be explained better and thus have added the necessary clarification.

**Resulting changes in manuscript**

Added an explanation regarding the notation of the candidate datasets (*p.10, lines 245-247* ).

**1.2.9 Comment 1.255**

> Figure 5 is referenced before Figure 4.

**Response**

That is correct. We changed the ordering of the figure.

**Resulting changes in manuscript**

The modified sentence in *p.11, lines 267-269*, correctly refers *Figure 4* before *Figure 5* now.

**1.2.10 Comment 1.264**

> Further explain and justify usage of Gaussian emulator.

**Response**

Thank you for pointing this out. We agree that both the explanation of the Gaussian process (GP) emulator and the justification of its use required further elaboration. In the revised manuscript, we have therefore expanded *Section 2.4.1* to (i) explain why we chose GP over other surrogate modeling approaches, specifically its ability to provide probabilistic predictions with uncertainty estimates and its superior performance with limited training data; (ii) introduce the mathematical definition of a GP; and (iii) clarify how the GP emulator is incorporated into our Bayesian data selection workflow.

**Resulting changes in manuscript**

The modified text from *Section 2.4.1* (*p. 11-12, lines 272 - 303 ff.* ) is given below.

Surrogate modeling serves as a computational enabler to overcome computational bottlenecks in calculating the KL divergence. As illustrated in Equation (10), calculating KL divergence requires posterior distributions, which are approximated through MCMC sampling (see Section 2.2). The accuracy of this approximation is based on the ability of MCMC chains to effectively explore the posterior space, which typically requires a large number of samples. Each sample necessitates evaluating the likelihood function, entailing one complete execution of the computational model. Therefore, this approach becomes infeasible for computationally expensive models. To tackle this challenge, we employ Gaussian process (GP) emulators, a non-intrusive surrogate modelling technique extensively used for reducing computational costs in Bayesian calibration workflows (Zhao and Kowalski, 2022). The widespread adoption of GP emulators stems from their ability to provide probabilistic predictions, allowing for rigorous quantification of the uncertainty associated with predictions. Furthermore, they offer efficient performance with limited training datasets compared to other machine learning approaches. Mathematically, a GP is defined by a mean function $m(\mathrm{x})$ and a covariance function $k(\mathrm{x}, \mathrm{x}')$ as shown in Equation (13).

$$f(\mathrm{x}) \sim \mathcal{GP}(m(\mathrm{x}), k(\mathrm{x}, \mathrm{x}')) \tag{13}$$

Both the mean and covariance functions in Equation (13) are parameterized by hyperparameters that are inferred from the training data. To implement the GP emulator, we utilize the `emulator` module of `PSimPy`, which harnesses RobustGaSP, an R package for Gaussian stochastic process emulation that provides robust estimates of the hyperparameters leading to enhanced predictive performance (Gu and Berger, 2016).

Figure 4 illustrates the key steps involved in the surrogate modeling phase. We start by generating a set of input parameters using the `sampler` module of `PSimPy`, which leverages space-filling schemes like Latin hypercube sampling. Next, we employ the `simulator` module to evaluate our computational model ($\mathcal{M}$), at these input points. The resulting model outputs are then post-processed according to the emulation strategy determined by the observation dimensionality. For scalar observations, we emulate the parameter-to-observable map, replacing the forward model $\mathcal{M}$ in Equation (9) with a cost-effective surrogate $\widehat{\mathcal{M}}$. Alternatively, for high-dimensional observations (e.g., velocity or position time series), we directly emulate the likelihood function, which quantifies the mismatch between model output and observation. This approach leverages the fact that the likelihood is scalar-valued regardless of observation dimensionality, thereby avoiding the computational challenges of constructing GP emulators with high-dimensional outputs. These post-processed outputs, together with the set of input points, constitute our training data used to build and train the GP emulator. We then validate the trained surrogate using k-fold cross-validation. After successful validation, the surrogate model is available for predictions.

**1.2.11  Comment 1.270**

> PsimPy is written in typewriter-like font here but in normal font earlier. Please be consistent.

**Response**

That is correct. We changed all instances of PSimPy to typewriter font `PSimPy` to be consistent.

**Resulting changes in manuscript**

We have corrected the font of `PSimPy` in all instances.

**1.2.12  Comment 1.284**

> Section 2.4.3 (Data selection) describes the process of computing the KL divergence for all combinations of observations and parameters. However, nothing is written about selecting data (or observations) based on the resulting matrix shown in Fig. 4. This needs to be described.

**Response**

Thank you for this comment. We agree that the use of the KL-divergence matrix (*Figure 5*) for selecting informative observations can be improved. In the revised manuscript, we have rewritten the paragraph in *Section 2.4.3* to explicitly explain how, for each parameter, the KL-divergence matrix is used to identify and select the most informative observation(s).

**Resulting changes in manuscript**

The modified text from *Section 2.4.3* (*p.13, lines 319 - 324 ff.* ) is given below.

> We marginalize the posterior distributions from the Bayesian calibration phase with respect to the parameters and compute KL divergence, as presented in Section 2.3. By calibrating $m$ parameters using $n$ observations, we end up with $n \times m$ KL divergence matrix, as shown in Figure 5, where each entry, $D_{\mathrm{KL}}^{i,j}$, quantifies the information that $i^{\mathrm{th}}$ observation $y_i$ provides for calibrating $j^{\mathrm{th}}$ parameter $\theta_j$. Using this KL divergence matrix we can identify the most informative observation $y_j^*$ for a given parameter $\theta_j$ by maximizing the KL divergence across available observations (as defined in Equation (12)).

**1.2.13  Comment 1.316**

> Subscripts that are not variables or indices should not be italic but normal font (i.e. $F_{\mathrm{res}}$). Furthermore, it is inconvenient to use $\theta$ for the angle here, since it is used as variable for the parameters earlier in the manuscript.

**Response**

Thank you for pointing this. We agree that variables and indices should not be italic and have corrected it. Additionally we agree that the usage of $\theta$ for slope angle could be confusing, since it is already used for parameters. Thus we have swapped $\theta$ with $\beta$ to represent slope angle.

**Resulting changes in manuscript**

We have modified *Equation 14* and *Equation 15* (Equation 13 and Equation 14 in the original manuscript) to make the subscripts non italic and to represent slope angle with $\beta$ instead of $\theta$.

**1.2.14   Comment 1.360**

> Experiment numbers (1-9) do not match numbers in table.

**Response**

Thank you for raising this point. We agree that the numbering of experiments in the text was not clearly aligned with the table, which could confuse readers. The confusion arose because *Subsection 4.7* discusses the results of both experiments 7 and 8 together. To address this, we have revised the subsection headings (4.1-4.7) to explicitly indicate which experiment(s) are discussed in each subsection. This provides a clear link between the table, the text, and the results presented.

**Resulting changes in manuscript**

To address the reviewer feedback we have modified the headings for subsections (4.1-4.7). The modified subsection headings are in *lines 402, 411, 418, 424, 430, 445, 451*.

**1.2.15   Comment 1.361**

> Italic subscripts that are not variables or indices.

**Response**

We thank the reviewer for this remark and would appreciate further clarification. The relevant passage in the submitted manuscript is:

> This section includes results for the curated set of numerical experiments discussed in Section 3.3. We adopt the workflow presented in Section 2.4 and the associated data: (i) Training dataset including set of sampled parameters and the corresponding

In this portion of the text, we could not identify any occurrences of italic subscripts that are non-variables or labels. We have also systematically reviewed and corrected all subscript notation throughout the manuscript. Could you kindly specify which expression at line 361 you are referring to?

**1.2.16   Comment 1.364**

> Figure 6: Some information are missing / not printed (at least in the provided pdf file): legend incomplete (no text next to lines); no axis labels; unit of $\xi$ not printed: Same issues with all the other figures of prior and posterior distributions. When it is unclear which line represents which quantity, it is nearly impossible to estimate the information content and correctness of the figures! This especially difficult for figure 11.

**Response**

Thank you for identifying these specific missing elements. As noted in our response to General Comment 5, these errors were caused by a PDF rendering error in the initial preprint version. We have verified that in the corrected manuscript, all figures displaying prior and posterior distributions (including *Figures 7 and 12*, which were Figures 6 and 11 in the original manuscript) are now rendered with complete axis labels, legends, and units. We apologize for the difficulty this caused in interpreting the results.

**Resulting changes in manuscript**

No changes to the manuscript content were necessary. The issue was resolved by correcting the rendering of the existing figures, which now display all axis labels, legends, and annotations as intended.

**1.2.17   Comment 1.366**

Subscript "max" should not be italic.

**Response**

Thank you for catching this. We fixed it and corrected all instances of such error.

**Resulting changes in manuscript**

We have modified the sentence (*p.19, line 403* ), such that the subscript "max" is not italic.

**1.2.18   Comment 1.368**

Are six significant digits for $\xi$ justified here?

**Response**

Thanks for pointing this out. We have omitted the additional decimal.

**Resulting changes in manuscript**

The modified sentence(*p.19, lines 405*), now omits the additional decimals for $\xi$. We made similar correction for $\xi$ in *Table 1*.

**1.2.19   Comment 1.372**

Where are the KL divergence values given / shown? reference Table 2

**Response**

Thank you. We have corrected this error.

**Resulting changes in manuscript**

The modified sentence (*p.19, line 409*), now refers to *Table 2*.

**1.2.20   Comment 1.387**

"x(t)" here is just text, not typeset as a formula. Please be consistent.

**Response**

Thanks for catching this, it has been corrected.

**Resulting changes in manuscript**

We have corrected the typeset error, refer (*p.22, line 424*).

**1.2.21 Comment 1.393**

> Figure 10: axis labels are missing. Are these examples taken from a set of 100 experiments? If so please state that fact. Otherwise the reader wonders how from 6 values you get such a smooth curve in figure 11a.

**Response**

Thank you for the comment.

- **Regarding missing labels:** As noted in our response to General Comment 5, these errors were caused by a PDF rendering error in the initial preprint version. We have verified that in the corrected manuscript, all figures displaying prior and posterior distributions (including *Figure 11*, which was Figure 10 in original manuscript) are now rendered with complete axis labels, legends, and units. We apologize for the difficulty this caused in interpreting the results.

- **Regarding Figure 10 (*Figure 11* in the revised manuscript):** We agree with your second comment and have added a sentence to explicitly state that the posterior distributions plotted in *Figure 11* (Figure 10 in the original manuscript), are selected from a set of 100 posterior distributions.

**Resulting changes in manuscript**

To address the reviewers feedback, we have modified the relevant text as (*p.23, lines 430-432*):

> We calibrated the friction parameters with 100 different datasets of velocity time series, each containing a varying number of time steps. Seven selected posterior distribution of $\xi$ from these calibrations is depicted in Figure 11, with the y-axis listing the number of velocity time steps used in the calibration.

**1.2.22 Comment 1.396**

> How is it visible that initial time steps have greater information content? That does not become clear. You show graphs for varying numbers of time steps but do not compare different onsets of the time series. Please clarify!

**Response**

Thank you for this comment. We recognize the ambiguity in our original phrasing. Our experiments used cumulative time windows (0-32, 0-64, ..., 0-1200), all starting from t=0. From *Figure 11* (Figure 10 in the original manuscript), we observed that the posterior change when adding early time steps (e.g., from 32 to 64) is considerably larger than when adding later time steps (e.g., from 1100 to 1200). This observation reflects diminishing marginal returns as more data is incorporated, but it does not directly address whether initial time steps inherently contain greater information content.

Your comment prompted us to investigate this question more directly. We performed additional calibration experiments using fixed-length time windows at different temporal positions: (0-100), (100-200), (200-300), (300-400), (700-800), (1000-1100), (1100-1200), and (1200-1300). These experiments directly compare information content across different portions of the time series while controlling for the amount of data used. The results (shown here in Figure 1) clearly demonstrate that posteriors become progressively wider for later time windows, confirming that early time steps contain greater information content than later ones.

**Resulting changes in manuscript**

Currently we intend to add Figure 1 in supplementary material to support the main narrative without disrupting the flow of the primary results.

**1.2.23 Comment 1.399**

> better: "number of velocity time steps". Otherwise it could be interpreted as KL divergence vs. specific time steps. Same in next sentence on line 400: better "Increasing number of time steps..."

**Response**

Thanks for your suggestion. We agree with it and have corrected accordingly.

**Resulting changes in manuscript**

The modified sentence now reads (*p.23, line 437-438*):

> Figure 12a, plots the variation of the KL divergence of the posterior and prior distributions of $\xi$ with the number of velocity time steps used for calibration. Increasing number of time steps resulted in higher KL divergence, indicating a positive correlation between the information gain and the number of time steps.

**1.2.24 Comment 1.364**

> Figure 11a: What is the reason for the shape of the KL div curve? Why is there a plateau between 300 and 500 steps and then an increase again?
>
> I suppose figure 11 shows number of time steps used (a) and time steps (b) but that's a guess since x-axis labels are missing. If that is so the start of the plateau corresponds to about 200 steps. It would be interesting to see that length also in figure 10. In figure 10 the three uppermost curves (1000, 1100, 1200 steps) look very similar. In figure 11a, however, the KL divergence is still increasing significantly in that range. Why is that?

**Response**

Thanks for the comment.

- **Regarding the shape of KL divergence curve:** We are conducting additional simulations to address this comment comprehensively. The results and revised analysis will be included in our resubmission.

- **Regarding Figure 11 (*Figure 12* in the revised manuscript):** As per your suggestion in comment 399, we have updated the axis label to explicitly denote the number of time steps used for calibration, improving clarity of the temporal scope.

- **Regarding Figure 10 (*Figure 11* in the revised manuscript):** Following your suggestion, we have added a posterior distribution calibrated with 200 velocity time steps.

- **Regarding the three uppermost curves:** We are conducting additional simulations to address this comment comprehensively. The results and revised analysis will be included in our resubmission.

**Resulting changes in manuscript**

The axis label in *Figure 12a, in p.24* (Figure 11a in the original manuscript), has been updated to denote the number of time steps. An additional posterior distribution (200 time steps) has been added to *Figure 11, in p.23* (Figure 10 in the original manuscript).

**1.2.25 Comment 1.410**

> Figure 12 is missing axis-labels again. A plot of KL divergence vs. time step would be informative, too. Which time window is used here for the observations?

**Response**

Thank you for the comment.

- **Regarding missing labels:** As noted in our response to General Comment 5, these errors were caused by a PDF rendering error in the initial preprint version. We have verified that in the corrected manuscript, all figures displaying prior and posterior distributions (including *Figure 13*, which was Figure 12 in the original manuscript) are now rendered with complete axis labels, legends, and units. We apologize for the difficulty this caused in interpreting the results.

- **Regarding KL divergence vs time step size plot:** Figure 11a, in the original manuscript represents a plot of KL divergence vs number of time steps. Could you clarify if you were suggesting a plot of KL divergence vs time step size?

- **Regarding the time window:** In the experiments presented in *Figure 13* (Figure 12 in the original manuscript), the observational datasets consist of the complete velocity time series. We vary the temporal resolution by adjusting the time step size, while using the complete time series.

**Resulting changes in manuscript**

No changes were necessary in the manuscript.

**1.2.26 Comment 1.413/414**

> you refer to section 2.1 where the discrepancy, i.e. measurement error is called $\varepsilon$. Here you write $\sigma$ which seems to refer to the STD of applied noise as given in Table 1. Be consistent with naming scheme!

**Response**

Thank you for pointing out this potential confusion. Both $\varepsilon$ and $\sigma$ are related but refer to different quantities: $\varepsilon$ is the discrepancy random variable itself, while $\sigma$ is the parameter (standard deviation) that characterizes its distribution.

As stated in *Section 2.1 (p.6, lines 166-168)*, we model the discrepancy $\varepsilon$ between measurements $y$ and model predictions $\mathcal{M}(s; \theta)$ as additive Gaussian noise: $\varepsilon \sim \mathcal{N}(0, \Sigma)$, where $\Sigma$ is the covariance matrix. For scalar observations, this reduces to $\varepsilon \sim \mathcal{N}(0, \sigma^2)$, where $\sigma$ is the standard deviation. For multiple independent observations, we use $\Sigma = \sigma^2 I$.

In experiments 1-6, we assume $\sigma$ based on heuristics. In experiments 7-8, we calibrate $\sigma$ alongside the friction coefficients. Since our synthetic measurements are generated by adding noise with known standard deviation (*Table 1*), successful calibration should recover these noise values, which is confirmed by the posterior distributions in *Figure 14 a-b* (Figure 13 a-b in the original manuscript).

**Resulting changes in manuscript**

To address the reviewer feedback we have included a clarification on the notation of $\sigma$ in *Section 2.1.1, p6, lines 166-172*. The modified text is given below.

> Typically, the noise is not known, such that we need to assume an ansatz, which constitutes our noise model. The additive Gaussian noise model is one of the most common noise models, where $\varepsilon$ is considered to be a realization of a Gaussian distribution, often with zero mean and a covariance matrix $\Sigma$, i.e. $\varepsilon \sim \mathcal{N}(0, \Sigma)$. The structure of the covariance matrix depends on the observational dataset we are calibrating. For scalar observations, the covariance matrix reduces to the variance $\sigma^2$. For multidimensional observations, we assume independent errors with constant standard deviation, leading to $\Sigma = \sigma^2 I$. We refer to $\sigma$ as the discrepancy parameter and in this study, it is either fixed based on heuristic assumptions or calibrated alongside the model parameters.

**1.2.27 Comment 1.512**

> Figure 11b shows the max. acceleration (assuming it is the blue line) at about 200 (no unit given). Figure 10 shows a narrow distribution only for 1000 steps or more. So how do you come to this optimality statement? That does not seem reasonable. There is not even a curve for 200 steps in figure 10.

**Response**

Thank you for this comment. We agree that the optimality claim requires further clarification.

- **Regarding Figure 11b (*Figure 12b* in the revised manuscript):** The acceleration is represented by the red line, while the blue line corresponds to velocity. The sliding mass reaches its maximum velocity at timestep 134 to be exact.

- **Regarding optimality:** We recognize this term was ambiguous. Our statement refers to the observation window that yields the maximum rate of information gain per time step, not the window that produces the most accurate final calibration (which, as you correctly note from *Figure 11* (Figure 10 in the original manuscript), requires $\sim$1000 time steps). Specifically, we observe that the rate of information gain—measured by the slope of the KL divergence in *Figure 12a* (Figure 11a in the original manuscript)—is steepest in the window from t=0 to approximately t=134, which coincides with the time required for the sliding mass to reach maximum velocity. Similarly, the coefficient of variation for the posteriors drops most sharply in this same window before flattening, as shown here in Figure 2. Furthermore, Figure 1 shows that the (0-100) time window produces the most contracted posterior among all fixed-length windows tested.

**Resulting changes in manuscript**

We have added clarification regarding the optimality statement in *p.30, line 566*.

**2   Changelog of additional manuscript changes**

We also report the following notable modifications. Changes are traceable in the attached diff.pdf file.

1. made the usage of abbreviation of Gaussian process (GP) consistent though out the manuscript.

2. swapped the index of observations $y$ with the index of parameters $\theta$ to be consistent with the matrix notation in *Figure 5*.

3. non italicized the subscripts through out the manuscript.

4. added additional funding details.

[Figure]

Figure 1: Posterior distributions of turbulent friction coefficient $\xi$ calibrated with a fixed observation window of 100 time steps, with varying onset time. Initial time steps provide the highest information gain.

[Figure]

Figure 2: Coefficient of variation (COV) of the posterior distribution of $\xi$ as a function of the number of velocity time steps used for calibration. The COV decreases sharply during the phase when the sliding mass reaches maximum velocity, then plateaus, indicating diminishing posterior contraction as additional observations are assimilated.

[Figure]

Figure 3: Vertical cross-section of the synthetic topographic model used in this study, with elevation (z) plotted against the horizontal coordinate (x). The topography is constant along the transverse direction. The red marker denotes the position of the initial release point projected onto this cross-section.

**References**

H. Zhao and J. Kowalski. Bayesian active learning for parameter calibration of landslide run-out models. *Landslides*, 19:2033–2045, 2022. doi: 10.1007/s10346-022-01857-z.

J. Aaron, S. McDougall, J. Kowalski, A. Mitchell, and N. Nolde. Probabilistic prediction of rock avalanche runout using a numerical model. *Landslides*, 19:2853–2869, 2022. doi: 10.1007/s10346-022-01939-y.

Holly A. Huber, Senta K. Georgia, and Stacey D. Finley. Systematic bayesian posterior analysis guided by kullback-leibler divergence facilitates hypothesis formation. *Journal of Theoretical Biology*, 558: 111341, 2023. ISSN 0022-5193. doi: https://doi.org/10.1016/j.jtbi.2022.111341. URL `https://www.sciencedirect.com/science/article/pii/S0022519322003320`.

Abhijit Chowdhary, Shanyin Tong, Georg Stadler, and Alen Alexanderian. Sensitivity analysis of the information gain in infinite-dimensional bayesian linear inverse problems. *International Journal for Uncertainty Quantification*, 14(6):17–35, 2024. ISSN 2152-5080. doi: 10.1615/int.j.uncertaintyquantification.2024051416. URL `http://dx.doi.org/10.1615/Int.J.UncertaintyQuantification.2024051416`.

---

## Author Comment (AC3)

**Revision letter**: response to Reviewer #2**

**egusphere-2025-4531**
**Bayesian data selection to quantify the value of data for landslide runout calibration**

January 26, 2026

We would like to thank Reviewer #2 for their valuable comments and suggestions, which improved the quality of the manuscript substantially. We present herein our responses to the comments of Reviewer #2.

**Italicized** line, figure, section numbers refer **to the revised manuscript**.

The changes in the revised manuscript with respect to the original manuscript can be traced in the `diff.pdf` file. Note that the line numbers in this file deviate from the pure manuscript PDF due to the tracked changes displayed.
* * *
Legend

> Shows original reviewer comment

**Response**   Contains direct response and explanations.

**Resulting changes in manuscript**   Summarizes resulting changes in the paper.

> Contains a direct quote of new or modified manuscript content with significant changes with respect to the original manuscript.

**1 Comments of Reviewer #2**

**1.1 Comment 1**

> Please define "calibration" as it has different definitions. For example, see `https://en.wikipedia.org/wiki/Calibration_(statistics)` and `https://doi.org/10.1214/23-BA1404`

**Response**

Thank you for this comment. We use the term "calibration" to refer to the process of inferring model parameters from observational data within a Bayesian framework. We have now added an explicit definition in Section 1 (Introduction) to avoid ambiguity.

**Resulting changes in manuscript**

To address the reviewer's feedback we have added the definition of calibration (*p.2 lines 33-34*).

**1.2 Comment 2**

> I'm also concerned with use of KL-divergence. I don't think author's rebuttal on this is sufficient. As the authors have used only uniform prior, the measure used is equivalent to entropy. Entropy is sensible measure for sharpness of the distribution. Instead of talking about KL-divergence, I suggest the authors would talk about entropy and use entropy also in case of non-uniform priors. Using entropy would focus on maximizing the sharpness of the posterior, while using KL can lead also maximal shift of the posterior.

**Response**

Thank you for bringing this to our attention. We agree that in the context of this study, where we use uniform priors, KL divergence between the posterior and the prior is equivalent to the entropy of the posterior. Accordingly, we have modified the method subsection 2.3 (now titled "Data selection using information-theoretic metrics") to mathematically demonstrate this equivalence and explicitly acknowledge it in the context of our work.

Our primary objective in this work is to quantify the information gained during calibration with a given observation. Based on our literature review, KL divergence between the posterior and prior is commonly used to measure the information gain during calibration[Haeusel et al., 2026, Chowdhary et al., 2024, Huber et al., 2023, Baptista et al., 2022]. Thus, we adopted the KL divergence framework to follow the standard convention and facilitate comparison with existing literature. For the landslide models under consideration, we typically have weakly informative priors for the parameters, since they are conceptual parameters that cannot be physically measured. In this context, maximizing KL divergence between posterior and prior is equivalent to minimizing the entropy of the posterior distribution.

Additionally, we have modified the discussion section to address the relative merits of entropy versus KL divergence for cases with non-uniform priors. In such settings, entropy can be a more suitable metric when posterior shifts are potentially misleading, since it focuses solely on posterior sharpness. In contrast, KL divergence is preferable when posterior shifts are informative, as it captures both posterior concentration and shift. However, both these metrics have limitations under model misspecification and prior-data conflict; thus, we recommend using posterior predictive validation as a complementary assessment tool.

**Resulting changes in manuscript**

To address the reviewer feedback we have modified *Section 2.3 (p. 9, lines 233-260)*. We introduced entropy and explicitly state its mathematical equivalence to KL divergence under uniform priors, and we added citations showing that KL divergence is the standard metric in the literature for quantifying information gain.

**2.3 Data selection using information-theoretic metrics**

In the Bayesian framework, our prior beliefs about the parameters $\theta$ are updated by observations, but the extent of this update varies depending on which observations are used. To quantify how informative each candidate observation is, we measure the information gained during calibration with observation $y$. Observations reduce parameter uncertainty by updating the prior distribution $P(\theta)$ to the posterior distribution $P(\theta \mid y)$. Thus, the information gained during this updating process corresponds to the reduced uncertainty. A fundamental information-theoretic measure of uncertainty is the entropy. For a random variable $\theta$ with probability distribution $P(\theta)$, entropy $H(\theta)$ is given as:

$$H(\theta) = - \int P(\theta) \log P(\theta) \, d\theta \tag{1}$$

Higher entropy indicates greater uncertainty about the parameter. Therefore, the change in entropy from prior to posterior quantifies the information gained from an observation. A standard measure to quantify information gain based on change in entropy is KL divergence [Kullback and Leibler, 1951], also known as relative entropy. In the Bayesian setting, KL divergence between the posterior distribution ($P(\theta \mid y)$) and the prior distribution ($P(\theta)$), as defined in Equation (2), admits a direct interpretation as information gain, since it quantifies the reduction in uncertainty about parameter $\theta$ achieved by conditioning on observation $y$. Higher KL divergence indicates greater reduction in uncertainty from prior to posterior, implying that the observation provides greater information about the parameters. Accordingly, we use this quantity as a criterion for data selection, consistent with its standard use in Bayesian optimal experimental design and inverse problems [Haeusel et al., 2026, Chowdhary et al., 2024, Huber et al., 2023, Baptista et al., 2022].

$$D_{\mathrm{KL}}(P(\theta \mid y) \,\|\, P(\theta)) = \int P(\theta \mid y) \log \left( \frac{P(\theta \mid y)}{P(\theta)} \right) \, d\theta \tag{2}$$

In the context of our work, we focus on calibrating landslide runout models with conceptual parameters that cannot be measured physically. For these parameters, we have limited prior information such as physically plausible bounds derived from literature and domain expertise. We therefore use uniform priors within these bounds, following standard practice in landslide runout calibration [Aaron et al., 2019, Navarro et al., 2018, Moretti et al., 2017]. For uniform priors, the prior entropy $H(P(\theta))$ is constant, so maximizing KL divergence is equivalent to minimizing the posterior entropy $H(P(\theta|y))$ (see Appendix A for derivation). In our numerical experiments with uniform priors, maximizing KL divergence is therefore equivalent to selecting observations that minimize posterior entropy, yielding the sharpest and most informative posterior distributions.

Additionally we included the limitation on the use of KL divergence as measure of improved parameter calibration in the presence of model misspecification and prior-data conflict, particularly when using non-uniform priors, (*p. 30, lines 551-559*).

Second, when model misspecification or prior-data conflict occurs, posteriors can contract to incorrect regions of parameter space, where higher KL divergence does not necessarily indicate better calibration. This limitation is exacerbated when using non-uniform priors. For non-uniform priors, KL divergence between the posterior and the prior no longer reduces to posterior entropy alone, as it captures both the sharpness of the posterior and its shift from the prior distribution. This introduces a trade-off: when a posterior contracts to an incorrect region, KL divergence can assign high information gain to a misleading result because it rewards both concentration and shift. Conversely, entropy focuses solely on posterior sharpness and does not account for whether the posterior has shifted toward more plausible parameter regions. When applying this framework in such scenarios, practitioners should therefore complement these information-theoretic metrics with robust validation methods such as posterior predictive checks.

**1.3 Comment 3**

> I found the Figure 2 and 3 schematics confusing, as by first look it looks like information flows from priors to likelihoods, which doesn't make sense. I don't have good suggestion how to change them, but wanted to mention this if the authors would have other ideas.

**Response**

Thank you for pointing this out. We agree that current schematics in Figure 2 and 3 could be misinterpreted as showing information flowing from the prior to the likelihood. These figures are intended to illustrate the Bayesian updating process, where the prior distribution is updated by the observational data through the likelihood function. To clarify this relationship, we have explicitly added a "Bayesian Updating" box that takes the prior and likelihood as inputs and produces the posterior. The revised figures are reproduced here as fig. 1 and fig. 2 for your convenience.

**Resulting changes in manuscript**

To address the reviewer's comment we have added the revised figures to the manuscript.

**1.4 Comment 4**

> The author's mention which MCMC algorithm is used, so the authors could also mention which convergence diagnostics were used from ArviZ package.

**Response**

Thank you for the suggestion. To assess the convergence of the MCMC chains, we relied on both qualitative and quantitative diagnostics from the ArviZ package. Qualitatively, we examined trace plots of the MCMC chains to check for proper mixing and to ensure the chains adequately explored the parameter space. Quantitatively, we used the Gelman-Rubin statistic (also called the potential scale reduction factor, $\hat{R}$), which assesses whether multiple chains have converged to the same distribution by comparing the variance between chains to the variance within chains. We adopted the standard threshold of $\hat{R} < 1.01$ to confirm adequate convergence as proposed by Vehtari et al. [2021]. The traces plots for selected experiments are provided here for reference as figs. 3 to 8 and the corresponding $\hat{R}$-values are reported in table 1. The $\hat{R}$ values reported in table 1 satisfy the convergence criterion $\hat{R} < 1.01$ for all but two cases, which attained $\hat{R} = 1.01$ and $\hat{R} = 1.011$ respectively. For these marginal cases, we found that the effective sample size (ESS) was adequate, extending the chain length reduced both $\hat{R}$ values below the 1.01 threshold, confirming convergence.

**Resulting changes in manuscript**

We have specified the diagnostics used to assess convergence of MCMC chains in the manuscript (*p. 13, lines 328-329*). Corresponding trace plots and the table containing $\hat{R}$-hat values are provided in the Appendix B of the revised manuscript.

**1.4.1 Comment 5**

> Explicitly define the prior in case study, now it seems based on the plots that it's uniform, but the priors were not explicitly defined in the text. This make huge difference for the use of KL, as it's then equivalent to entropy which measures just the sharpness. Without explicitly mentioning the prior in the text, it takes more time and effort from the reader to see what has been actually used.

**Response**

Thank you for spotting this. We agree that interpretation of KL divergence results changes based on the choice of prior. We have therefore revised Section 3.3 (Design of Numerical Experiments) to provide

a clear description of the prior distributions used in the case study. As you correctly noted, we employ uniform priors, which we have now explicitly stated along with appropriate justification and citations.

**Resulting changes in manuscript**

Added description of priors used in Section 3.3 (*p.18, lines 408-410*), repeated below for convenience.

> In all the 8 experiments we assume uniform priors for the friction parameters. The bounds for the priors are chosen from the literature as discussed in Section 2.3, for $\mu \in [0.02, 0.3]$ and $\xi \in [100, 2200]\ m/s^2$. Additionally in experiment 7 and 8, we assume an uniform prior for the discrepancy parameters $\sigma_{vel_{TS}}$ and $\sigma_{pos_{TS}}$ with bounds $[1, 5]$ and $[0, 100]$ respectively.

**2 Figures from manuscript**

[Figure]

Figure 1: Schematic illustration of the Bayesian inference process. The likelihood function acts as the core driver of updating the prior distribution to posterior distribution based on observational data. The resulting posterior depends strongly on type and quality of the observation.

[Figure]

Figure 2: Schematic illustration of the Bayesian data selection process using information-theoretic metrics. Multiple calibration routines are performed in parallel, each using the same likelihood function but leveraging different observations to update the prior distributions. By comparing each resulting posterior distribution against the prior, we can quantify the information gained during calibration relative to observations.

[Figure]

Figure 3: MCMC trace plots for (a) Dry Coulomb friction coefficient ($\mu$) and (b) Turbulent friction coefficient ($\xi$) based on calibration with maximum velocity ($U_{\max}$) as observation. Each color represents an independent chain. Left panels show kernel density estimates of the marginal posterior distributions. Right panels show trace plots across iterations, demonstrating chain convergence and adequate mixing.

[Figure]

Figure 4: MCMC trace plots for (a) Dry Coulomb friction coefficient ($\mu$) and (b) Turbulent friction coefficient ($\xi$) based on calibration with runout distance ($X_{\mathrm{end}}$) as observation. Each color represents an independent chain. Left panels show kernel density estimates of the marginal posterior distributions. Right panels show trace plots across iterations, demonstrating chain convergence and adequate mixing.

[Figure]

Figure 5: MCMC trace plots for (a) Dry Coulomb friction coefficient ($\mu$) and (b) Turbulent friction coefficient ($\xi$) based on calibration with velocity time series as observation. Each color represents an independent chain. Left panels show kernel density estimates of the marginal posterior distributions. Right panels show trace plots across iterations, demonstrating chain convergence and adequate mixing.

[Figure]

Figure 6: MCMC trace plots for (a) Dry Coulomb friction coefficient ($\mu$) and (b) Turbulent friction coefficient ($\xi$) based on calibration with position time series as observation. Each color represents an independent chain. Left panels show kernel density estimates of the marginal posterior distributions. Right panels show trace plots across iterations, demonstrating chain convergence and adequate mixing.

[Figure]

Figure 7: MCMC trace plots for (a) Dry Coulomb friction coefficient ($\mu$) and (b) Turbulent friction coefficient ($\xi$) (c) Velocity discrepancy parameter ($\sigma_{\mathrm{vel_{TS}}}$) based on calibration with velocity time series as observation. Each color represents an independent chain. Left panels show kernel density estimates of the marginal posterior distributions. Right panels show trace plots across iterations, demonstrating chain convergence and adequate mixing.

[Figure]

Figure 8: MCMC trace plots for (a) Dry Coulomb friction coefficient ($\mu$) and (b) Turbulent friction coefficient ($\xi$) (c) Position discrepancy parameter ($\sigma_{\mathrm{pos_{TS}}}$) based on calibration with position time series as observation. Each color represents an independent chain. Left panels show kernel density estimates of the marginal posterior distributions. Right panels show trace plots across iterations, demonstrating chain convergence and adequate mixing.

| Observations | R_hat | | |
|---|---|---|---|
| | Dry Coulomb friction coefficient | Turbulent friction coefficient | Discrepancy parameter |
| Maximum velocity $U_{\max}$ | 1.007 | 1.007 | – |
| runout distance $X_{\mathrm{end}}$ | 1.008 | 1.009 | – |
| Velocity time series u($t$) | 1.006 | 1.005 | – |
| Position time series x($t$) | 1.010 | 1.010 | – |
| Velocity time series u($t$) (discrepancy) | 1.007 | 1.008 | 1.011 |
| Position time series x($t$) (discrepancy) | 1.009 | 1.009 | 1.007 |

Table 1: Gelman-Rubin statistics ($\hat{R}$) for MCMC sampling of friction coefficients and the discrepancy parameter. $\hat{R}$ values indicate successful chain convergence for all parameters across different observations.

**References**

Lea J. Haeusel, Jonas Nitzler, Lea J. Kãűglmeier, and Wolfgang A. Wall. Multi-physics-enhanced bayesian inverse analysis: Information gain from additional fields. *Computer Methods in Applied Mechanics and Engineering*, 452:118735, 2026. ISSN 0045-7825. doi: https://doi.org/10.1016/j.cma.2026.118735. URL https://www.sciencedirect.com/science/article/pii/S0045782526000095.

Abhijit Chowdhary, Shanyin Tong, Georg Stadler, and Alen Alexanderian. Sensitivity analysis of the information gain in infinite-dimensional bayesian linear inverse problems. *International Journal for Uncertainty Quantification*, 14(6):17–35, 2024. ISSN 2152-5080. doi: 10.1615/int.j.uncertaintyquantification.2024051416. URL http://dx.doi.org/10.1615/Int.J.UncertaintyQuantification.2024051416.

Holly A. Huber, Senta K. Georgia, and Stacey D. Finley. Systematic bayesian posterior analysis guided by kullback-leibler divergence facilitates hypothesis formation. *Journal of Theoretical Biology*, 558: 111341, 2023. ISSN 0022-5193. doi: https://doi.org/10.1016/j.jtbi.2022.111341. URL https://www.sciencedirect.com/science/article/pii/S0022519322003320.

Ricardo Baptista, Lianghao Cao, Joshua Chen, Omar Ghattas, Fengyi Li, Youssef M. Marzouk, and J. Tinsley Oden. Bayesian model calibration for block copolymer self-assembly: Likelihood-free inference and expected information gain computation via measure transport, 2022. URL https://arxiv.org/abs/2206.11343.

S. Kullback and R. A. Leibler. On information and sufficiency. *Ann. Math. Stat.*, 22:79–86, 1951. URL https://api.semanticscholar.org/CorpusID:120349231.

J. Aaron, S. McDougall, and N. Nolde. Two methodologies to calibrate landslide runout models. *Landslides*, 16:907–920, 2019. doi: 10.1007/s10346-018-1116-8.

Maria Navarro, Olivier P. Le Maître, Ibrahim Hoteit, David L. George, Kyle T. Mandli, and Omar M. Knio. Surrogate-based parameter inference in debris flow model. *Computational Geosciences*, 22(6): 1447–1463, Dec 2018. ISSN 1573-1499. doi: 10.1007/s10596-018-9765-1. URL https://doi.org/10.1007/s10596-018-9765-1.

L. Moretti, A. Mangeney, F. Walter, Y. Capdeville, T. Bodin, E. Stutzmann, and A. Le Friant. Constraining landslide characteristics with bayesian inversion of field and seismic data. *Geophys. J. Int.*, 2020:1–15, 2017. doi: 10.1093/gji/ggaa056.

Aki Vehtari, Andrew Gelman, Daniel Simpson, Bob Carpenter, and Paul-Christian Bürkner. Rank-Normalization, Folding, and Localization: An Improved $\widehat{R}$ for Assessing Convergence of MCMC (with Discussion). *Bayesian Analysis*, 16(2):667 – 718, 2021. doi: 10.1214/20-BA1221. URL https://doi.org/10.1214/20-BA1221.

---

## Author Comment (AC4)

**Revision letter: response to Reviewer #1**

**egusphere-2025-4531**
**Bayesian data selection to quantify the value of data for landslide runout calibration**

January 26, 2026

We would like to thank Reviewer #1 for their valuable comments and suggestions, which improved the quality of the manuscript substantially. We present herein our responses to the comments of Reviewer #1.

**Italicized** line, figure, section numbers refer **to the revised manuscript**.

The changes in the revised manuscript with respect to the original manuscript can be traced in the `diff.pdf` file. Note that the line numbers in this file deviate from the pure manuscript PDF due to the tracked changes displayed.
* * *
**Legend**

| Shows original reviewer comment

**Response** Contains direct response and explanations.

**Resulting changes in manuscript** Summarizes resulting changes in the paper.

> Contains a direct quote of new or modified manuscript content with significant changes with respect to the original manuscript.

**1 Comments of Reviewer #1**

**1.1 Comment 1.364**

> Figure 11a: What is the reason for the shape of the KL div curve? Why is there a plateau between 300 and 500 steps and then an increase again?
>
> I suppose figure 11 shows number of time steps used (a) and time steps (b) but that's a guess since x-axis labels are missing. If that is so the start of the plateau corresponds to about 200 steps. It would be interesting to see that length also in figure 10. In figure 10 the three uppermost curves (1000, 1100, 1200 steps) look very similar. In figure 11a, however, the KL divergence is still increasing significantly in that range. Why is that?

**Response**

Thanks for the comment.

To address this comment, we recomputed the KL divergence between the posterior and prior distributions of the turbulent friction coefficient $\xi$ using datasets with finer spacing in the number of velocity time steps. Previously, we calibrated posteriors using 100 velocity time series datasets, each containing progressively more time steps with larger increments between successive datasets. We now use 1000 datasets with smaller increments in time step count between successive datasets, eliminating interpolation artifacts. The analysis confirms three distinct phases observed in the original figure: (1) a sharp rise as the mass accelerates, (2) a plateau where KL divergence remains relatively constant, and (3) a subsequent increase until saturation.

However, we identified an artifact in Figure 11(a) (*Figure 12(a)* in the revised manuscript): the emulator was trained on velocity time series data only up to the initial 1334 time steps, the maximum length available consistently across all training simulations. Although calibration used nominally higher number of time steps, observational data was effectively truncated to 1334 steps to match the emulator output dimensionality. This truncation was inadvertently overlooked during plotting. We have now corrected Figure 11(a) (reproduced here as fig. 1) to show KL divergence only up to 1334 time steps, accurately reflecting the actual calibration range.

- **Regarding the shape of KL divergence curve:**

  To understand the behavior of the KL divergence curve, we examine the coefficient of variation (COV) of the posterior distribution of $\xi$ versus the number of time steps, plotted in fig. 3, as a complementary measure of uncertainty. The COV drops sharply in the observation window corresponding to when the sliding mass accelerates and attains maximum velocity, then slowly decreases toward saturation. This indicates the broader trend that as velocity time series length increases, information gain diminishes, suggesting diminishing returns. While the rate of decrease in COV becomes modestly steeper beyond $\sim 500$ time steps, this change is not as pronounced as depicted in the KL divergence plot. This contrast arises from the logarithmic nature of KL divergence, which is more sensitive to small relative changes. When the COV is plotted on a logarithmic scale (fig. 4), its trend closely mirrors that of the KL divergence. This comparison indicates that the increase in KL divergence after $\sim 500$ time steps is more moderate than it seems.

  Nevertheless, there is an observable rise in KL divergence values after a plateau (approximately between 300 to 500 time steps). We hypothesize that this behavior reflects friction regime transitions described by Hergarten [2024]. During the initial acceleration phase with high velocity, velocity-dependent turbulent friction may dominate, causing a sharp rise in KL divergence for $\xi$. The plateau likely marks a transition zone where dynamics are relatively stable. As the mass decelerates and velocity decreases, Coulomb friction may become dominant, producing a second, more gradual increase. To explore this hypothesis, we compared KL divergence change with number of time steps for both $\mu$ and $\xi$ (see fig. 5). In the initial turbulent-friction dominated phase, $\xi$ increases sharply compared to $\mu$. This pattern reverses in the later Coulomb friction dominated phase where $\mu$ rises more steeply than $\xi$, with both parameters showing minimal change during the transition zone. This behavior is consistent with the regime-switching explanation.

- **Regarding Figure 11 (*Figure 12* in the revised manuscript):** As per your suggestion in comment 399, we have updated the axis label to explicitly denote the number of time steps used for calibration, improving clarity of the temporal scope.

- **Regarding Figure 10 (*Figure 11* in the revised manuscript):**

  Following your suggestion, we added the posterior distribution calibrated using the initial 200 velocity time steps. We found that the posterior distribution of $\xi$ based on these 200 time steps exhibits a trend of posterior contraction similar to that observed in the original figure. The revised figure is reproduced here for your convenience (fig. 2).

- **Regarding the three uppermost curves:**

  As shown in fig. 3, the coefficient of variation for the posterior distributions corresponding to the three uppermost curves (1000, 1100, 1200 time steps) in fig. 2 varies minimally, reflecting limited additional contraction. However, because KL divergence operates on a log scale, these small variations are amplified fig. 1, creating the appearance of significant increase despite minimal changes in the posterior distributions.

**Resulting changes in manuscript**

The x-axis label in *Figure 12a* (Figure 11a in the original manuscript) has been updated to indicate the number of time steps. Additionally, KL divergence values beyond 1334 time steps have been removed from this figure to accurately reflect the actual calibration range. An additional posterior distribution (200 time steps) has been added to *Figure 11* (Figure 10 in the original manuscript).

**2  Figures changed in the manuscript**

[Figure]

Figure 1: (a) Variation of Kullback-Leibler divergence of the posterior and prior distributions of turbulent friction coefficient $\xi$ with number of velocity time steps. (b) Variation of velocity and acceleration with number of time steps.

[Figure]

Figure 2: Posterior distributions of turbulent friction coefficient $\xi$ calibrated with a varying number of velocity time steps. The rate of information gain is more profound in the initial time steps than the later.

**3 Additional figures for explanation**

[Figure]

Figure 3: Coefficient of variation (COV) of the posterior distribution of $\xi$ as a function of the number of velocity time steps used for calibration. The COV decreases sharply during the phase when the sliding mass reaches maximum velocity, then plateaus, indicating diminishing posterior contraction as additional observations are assimilated.

[Figure]

Figure 4: Coefficient of variation (COV) and KL divergence of the posterior distribution of $\xi$ as a function of the number of velocity time steps used for calibration. When COV is plotted on a log scale, the variations are amplified similarly to the KL divergence curve, demonstrating the log-scale amplification effect.

[Figure]

Figure 5: KL divergence between posterior and prior distributions of $\mu$ and $\xi$ calibrated with varying number of velocity time steps.

**References**

S. Hergarten. Scaling between volume and runout of rock avalanches explained by a modified voellmy rheology. *Earth Surface Dynamics*, 12(1):219–229, 2024. doi: 10.5194/esurf-12-219-2024. URL `https://esurf.copernicus.org/articles/12/219/2024/`.